# Video swin-CLSTM transformer: Enhancing human action recognition with optical flow and long-term dependencies

Jun Qin[1,2©], Shenwei Chen[1,3©], Zheng Ye[1]*, Jing Liu[1,2], Zhou Liu[1,3]

1 College of Computer Science, South-Central Minzu University, Wuhan, Hubei, China, 2 Hubei Provincial Engineering Research Center for Intelligent Management of Manufacturing Enterprises, Wuhan, Hubei, China, 3 Hubei Provincial Engineering Research Center of Agricultural Blockchain and Intelligent Management, Wuhan, Hubei, China

© These authors contributed equally to this work.
* yezheng@scuec.edu.cn

## Abstract

As video data volumes soar exponentially, the significance of video content analysis, particularly Human Action Recognition (HAR), has become increasingly prominent in fields such as intelligent surveillance, sports analytics, medical rehabilitation, and virtual reality. However, current deep learning-based HAR methods encounter challenges in recognizing subtle actions within complex backgrounds, comprehending long-term semantics, and maintaining computational efficiency. To address these challenges, we introduce the Video Swin-CLSTM Transformer. Based on the Video Swin Transformer backbone, our model incorporates optical flow information at the input stage to effectively counteract background interference, employing a sparse sampling strategy. Combined with the backbone's 3D Patch Partition and Patch Merging techniques, it efficiently extracts and fuses multi-level features from both optical flow and raw RGB inputs, thereby enhancing the model's ability to capture motion characteristics in complex backgrounds. Additionally, by embedding Convolutional Long Short-Term Memory (ConvLSTM) units, the model's capacity to capture and understand long-term dependencies among key actions in videos is further enhanced. Experiments on the UCF-101 dataset demonstrate that our model achieves mean Top-1/Top-5 accuracies of 92.8% and 99.4%, which are 3.2% and 2.0% higher than those of the baseline model, while the computational cost is reduced by an average of 3.3% at peak performance compared to models without optical flow. Ablation studies further validate the effectiveness of our model's crucial components, with the integration of optical flow and the embedding of ConvLSTM modules yielding maximum improvements in mean Top-1 accuracy of 2.6% and 1.9%, respectively. Notably, employing our custom ImageNet-1K-LSTM pre-training model results in a maximum increase of 2.7% in mean Top-1 accuracy compared to traditional ImageNet-1K pre-training model. These experimental results indicate that

**Data availability statement:** We have uploaded the minimal data set of the experimental results to the GitHub repository. The link is: [https://github.com/cherr ycsw/Video_Swin_CLSTM/tree/main/minimal_data_set]. All other relevant data are within the manuscript and its Supporting Information files.

**Funding:** This work was supported by the Hubei Province Key Research and Development Special Project of Science and Technology Innovation Plan (No. 2023BAB087), the Wuhan Key Research and Development Projects (No. 2023010402010614), the Central Government Guides Local Funds for Science and Technology Development (No. ZYYD2024QY08) and the Wuhan Knowledge Innovation Special Dawn Project (No. 2023010201020465). Further support was provided by the open competition project for selecting the best candidates, Wuhan East Lake High-tech Development Zone (No. 2024KJB328) and the Fund for Research Platform of South-Central Minzu University (No. CZQ24011). There was no additional external funding received for this study.

**Competing interests:** The authors have declared that no competing interests exist.

our model offers certain advantages over other Swin Transformer-based methods for video HAR tasks.

## Introduction

The rapid growth and widespread application of video data have made video content analysis increasingly crucial in contemporary society. Fields such as intelligent surveillance, sports analytics, medical rehabilitation, and virtual reality demand more accurate and efficient recognition of human actions in videos. In intelligent surveillance, promptly detecting abnormal behaviors like falls or sudden running enhances the real-time responsiveness of monitoring systems, helping to prevent safety incidents [1]. In sports analytics, recognizing athletes' movements and actions with precision is essential for performance evaluation, injury prevention, and strategic planning. It offers coaches and teams data-driven insights to optimize training and in-game decisions [2]. In medical rehabilitation, accurately identifying patients' movements aids in developing personalized rehabilitation plans, thereby improving recovery outcomes [3]. Similarly, in virtual reality, the quality of natural interactions is contingent upon the accurate capture and recognition of user actions.

Despite notable advancements in deep learning-based Human Action Recognition (HAR), several challenges persist, especially in recognizing subtle actions within complex backgrounds, reducing computational costs, and capturing long-term temporal dependencies [4]. Traditional deep learning methods primarily analyze individual video frames, lacking the continuous spatiotemporal information required for robust recognition [5]. Moreover, factors like variations in human appearance, cluttered backgrounds, and changing camera angles often obscure human silhouettes, complicating the model's ability to distinguish subjects from their environment [6]. This limitation hampers the recognition of key body parts or micro-movements, which are essential for accurate HAR. To address these issues, many researchers have opted for computationally intensive models, sacrificing efficiency for improved accuracy [7]. However, balancing computational complexity with recognition performance remains a significant hurdle.

Another critical challenge lies in capturing long-term semantic information. Complex actions often consist of multiple sub-actions, each with distinct semantic significance. Effective recognition requires analyzing these sub-actions over extended periods to uncover meaningful temporal relationships [8]. Existing approaches [1,3,9–13] primarily fuse spatial and temporal information directly but often fail to maintain context over longer durations, leading to fragmented understanding and missed dynamic changes. These limitations hinder performance in real-world applications where action continuity and variability are crucial.

To address these challenges, this study proposes the Video Swin-CLSTM Transformer, a novel architecture that integrates the spatial feature extraction capabilities of the Video Swin Transformer [13] with the temporal modeling strength of ConvLSTM units [8]. The model leverages optical flow information, which captures motion

dynamics between frames by analyzing pixel value changes over time. Optical flow's appearance invariance minimizes the impact of background variations, enabling the model to effectively capture instantaneous micro-motions [14]. By integrating optical flow with spatial features, the model addresses dimensional mismatch and information desynchronization, providing crucial temporal context often absent in traditional methods.

Additionally, the use of ConvLSTM units enhances the model's ability to understand long-term semantics by capturing both spatial and temporal dependencies. This capability is essential for handling complex actions involving multiple sub-actions over extended periods. The model also incorporates the Swin Transformer [15] architecture along with 3D Patch Partition and Patch Merging techniques to optimize spatiotemporal feature extraction. By potentially downsampling optical flow data instead of processing full RGB frames, the model reduces computational overhead while preserving relevant motion features.

In summary, this study makes three key contributions:

1. Introducing the Video Swin-CLSTM Transformer, a novel architecture that integrates optical flow and spatial features to enhance the recognition of subtle actions in complex backgrounds.

2. Incorporating ConvLSTM units to improve long-term semantic understanding, enabling the model to process extended video sequences effectively.

3. Utilizing sparse optical flow sampling alongside 3D Patch Partition and Patch Merging techniques within the Swin Transformer framework to balance computational efficiency with rich spatiotemporal feature extraction.

These advancements push the boundaries of HAR technology, broadening its applicability across various domains.

## Related work

Video-based HAR faces critical challenges such as subtle motion amid noise, long-term context, and the efficiency-accuracy trade-off. To address these issues, diverse methodologies have been explored, among which deep learning-based approaches have emerged as dominant solutions due to their ability to learn hierarchical spatiotemporal representations through end-to-end training. Unlike earlier methods [9–11] that relied on handcrafted features or shallow models, deep neural networks excel at capturing fine-grained motion dynamics and modeling long-range contextual relationships, significantly improving robustness and discriminative power. Current mainstream architectures fall into four categories: Two-stream Convolutional Neural Networks (CNNs), Recurrent Neural Networks (RNNs), 3D Convolutional Neural Networks (3D CNNs), and Transformer-based models.

### Based on two-stream CNNs

The two-stream CNNs framework has significantly improved HAR performance by integrating spatial and temporal information. Early two-stream architectures [1] relied on optical flow to capture motion cues, but their sensitivity degraded significantly under background noise (e.g., achieving 59.4% mean accuracy on HMDB51, with lower performance on fine-grained actions). Bilen et al. [16] addressed this by encoding motion patterns into noise-robust dynamic images, improving fine-grained action accuracy by 4.6% but led to a 6.2% performance loss on rapid motions due to temporal abstraction compared to the original SOTA method. While this enhanced noise resilience, it exacerbated the existing limitation of modeling actions beyond short snippets (<5s). Wang et al. [17] proposed Temporal Segment Networks (TSN) to address long-term temporal modeling by aggregating predictions from sparsely sampled video snippets. On HMDB51, TSN achieved 68.5% top-1 accuracy for procedural actions (e.g., "Brushing Hair") when trained with RGB+Flow input, surpassing single-snippet baselines by 9.4%. However, subsequent analysis revealed limitations in transient action detection: Feichtenhofer et al. [18] demonstrated that TSN's sparse sampling strategy increased miss rates by on average for short-duration actions like "sneezing" in Kinetics-400. This highlights a fundamental trade-off between capturing long-range

context and detecting abrupt actions. Feichtenhofer et al. [6] compressed motion semantics via cross-stream distillation, attaining 65.4% accuracy on HMDB51 with single-stream cost, but suffering 6.8% cross-domain degradation. These limitations, particularly fragmented temporal modeling and cross-domain degradation, motivate the integration of RNNs to capture action evolution through sequential hidden states.

### Based on RNNs

RNNs, owing to their recurrent connections within hidden layers, are inherently adept at analyzing sequential data. The foundational LRCN architecture [5], which cascades convolutional layers with Long Short-Term Memory (LSTM) units [19–21], achieved 74.3% accuracy on UCF101 for short-term actions like "typing". However, its dependency on optical flow inputs amplified noise interference in complex scenes, reducing " typing "recognition accuracy to 60.4% on UCF101 under cluttered backgrounds. To address noise sensitivity, Visual Attention LSTM [22] introduced spatial attention gates within LSTM cells, selectively enhancing motion-salient regions. This improved HMDB51 fine-grained accuracy by 3.7% (over pooled LSTMs) while increasing the model's parameters, underscoring the trade-off between robustness and efficiency. Hierarchical GRUs [23] attempted to resolve long-term dependencies by partitioning videos into temporal segments, achieving 2.1% accuracy gains on procedural tasks in UCF-101. However, the segmentation introduced fragmented temporal processing, significantly delaying the detection of transient actions (e.g., "falling"), which poses a fundamental limitation for real-time applications. Skip RNN [24] dynamically pruned 58% of redundant time steps via LSTM gating mechanisms, achieving 85 FPS inference on HMDB51 with a 4.3% accuracy degradation (from 57.1% to 52.8%). This approach relied on knowledge distillation from a Kinetics-400 pretrained teacher LSTM, limiting scalability to domains lacking large-scale annotated video data. RNNs struggle with sequential computation, limiting parallelization and fragmenting spatiotemporal learning. This drives the use of 3D CNNs, which unify spatial and temporal modeling through volumetric kernels, enabling parallelized joint learning of motion and context.

### Based on 3D CNNs

3D CNNs have significantly advanced HAR by effectively modeling spatial and temporal information in videos. The foundational C3D network [4] employed uniform 3D kernels to capture short-term motion patterns, achieving 85.2% accuracy on UCF101 for coarse actions like "running", but performed poorly on fine-grained motions due to background noise. Tran et al. [25] enhanced noise robustness in ResNet-3D by incorporating residual connections and spatial attention, boosting HMDB51 accuracy to 78.7% while maintaining 73.3% on Sports-1M. However, its fixed 16-frame inputs restricted temporal context to 1~2 seconds, limiting long-term action understanding. To address this, Carreira et al. [2] introduced I3D, inflating 2D ImageNet-pretrained kernels into 3D and achieving high accuracy on UCF101 by aggregating multi-scale temporal features. Despite its success, I3D's high computational cost (306 GFLOPS per clip) hindered real-time deployment. Efficiency-focused approaches like X3D [18] progressively expanded temporal and spatial capacities, reducing FLOPs by 74% while retaining 76% of I3D's accuracy on Kinetics-400. However, X3D's compressed temporal resolution (8-frame windows) reduced sensitivity to transient actions (e.g., an around 6.3% increase in miss rate for "sneezing"). These limitations of 3D CNNs—fixed kernels and high computational costs—have driven the adoption of Transformers, which leverage self-attention to dynamically model spatiotemporal relationships and long-range dependencies more efficiently.

### Based on transformer models

Transformer-based models have significantly advanced on HAR by effectively modeling global spatiotemporal dependencies through self-attention mechanisms. TimeSformer [26] pioneered this domain with decoupled spatial-temporal attention, achieving 80.7% accuracy on Kinetics-400 while reducing computational costs by 40% compared to 3D CNNs, though its rigid attention separation limited performance on fine-grained motions like "handwriting" (68.2% on

Something-Something V2). ViViT [27] addressed spatial-temporal interaction through tubelet embeddings, improving fine-grained action accuracy to 72.4% on the same benchmark, albeit with high computational demands (320 GFLOPS for 32-frame inputs). MViT [28] further balanced local-global information via hierarchical multiscale tokens, attaining 81.2% Kinetics-400 accuracy, though its fixed-scale tokenization struggled with abrupt actions like sneezing (73.1%). The Swin Transformer [15] revolutionized efficiency through sliding window attention, reducing computation by 30% versus ViViT, while Video Swin Transformer [13] extended this with 3D shifted windows to achieve 84.9% Kinetics-400 accuracy at 22 GFLOPS, though fragmented window processing caused 5.8% accuracy drops in long-range temporal reasoning tasks. VideoMAE [29] optimized scalability through self-supervised pretraining with 90% token masking, reaching 86.6% Kinetics-400 accuracy, though incurring 1.2s/clip inference latency from dense frame sampling. These models collectively demonstrate Transformer architectures' superiority in global dependency modeling and long-video processing over CNNs/RNNs, while maintaining efficiency through adaptive attention mechanisms.

A review of the existing literature indicates that Transformer-based approaches are highly effective for HAR, particularly in capturing spatiotemporal information. The Video Swin Transformer excels at balancing computational complexity with feature extraction capabilities. However, challenges remain in handling complex backgrounds and recognizing subtle actions. To address these challenges, this study proposes several innovations: first, by integrating optical flow and spatial information, we enhance HAR using an improved Video Swin Transformer, where optical flow captures subtle motion details and mitigates the impact of background variations; second, we optimize the model's architecture and sampling method to reduce computational complexity while preserving high accuracy; finally, by incorporating a ConvLSTM module, we enhance the model's ability to capture longer-term temporal dependencies, thereby improving recognition accuracy and robustness. These advancements provide new solutions for achieving efficient and accurate HAR in complex environments.

## Methods

### Overall architecture

The overall architecture of the proposed Video Swin-CLSTM Transformer is illustrated in Fig 1. The overall architecture comprises two main components: the input module and the computational module. And the final output of the model is

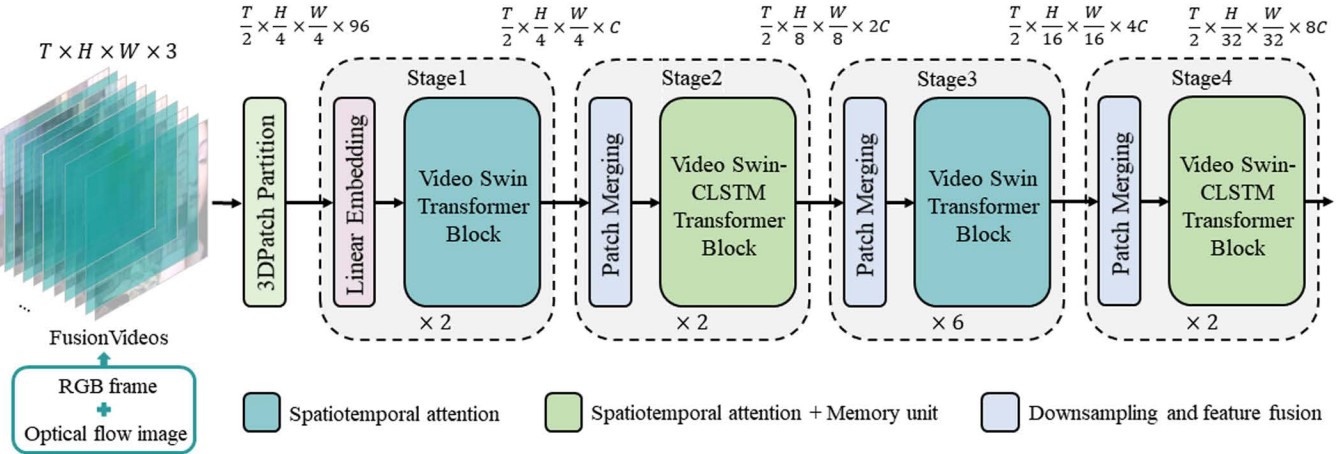

**Fig 1. The architecture of Video Swin-CLSTM Transformer.** It is composed of four stages, with the number of blocks in each stage being 2, 2, 6, and 2, respectively. In Stage 2 and Stage 4, the core modules are Video Swin-CLSTM Transformer blocks with spatiotemporal attention and memory units, while the remaining stages utilize Video Swin Transformer blocks with spatiotemporal attention only. The description above the figure illustrates the changes in the model data dimensions throughout the process.

classified by the classification header [27]. The input module processes a fused stream of RGB and optical flow images, while the computational module utilizes the tiny version of the Video Swin Transformer (Video Swin-T) [13] as its backbone, with the number of blocks in successive stages being (2, 2, 6, 2). The core modules in Stage 2 and Stage 4 are Video Swin-CLSTM Transformer blocks, while the core modules in the other stages are Video Swin Transformer blocks. Initially, the fused video input with dimensions T × H × W × 3 is fed into the 3D Patch Partition module, which segments the video into non-overlapping patches. Each patch covers a 2 × 4 × 4 region of adjacent pixels and is then flattened along the channel axis. Since each patch contains 32 pixels, each with three color channels (R, G, and B), the resulting feature dimension per patch is 96. Thus, the shape of the fused video transitions from [T, H, W, 3] to [T/2, H/4, W/4, 96] after 3D Patch Partition processing. In Stage 1, the Linear Embedding layer linearly transforms the channel data for each pixel, reducing the number of channels from 96 to C, thereby reshaping the video to [T/2, H/4, W/4, C]. In Stage 2, Patch Merging combines adjacent 2 × 2 patches, altering the shape to [T/2, H/8, W/8, 2C]. Patch Merging operations in Stages 3 and 4 follow a similar process, halving the height (H) and width (W) dimensions while doubling the channel count C. The 3D Patch Merging technique, mirroring the 2D approach found in the Swin Transformer, halves the spatial dimensions by a factor of two. It selectively picks elements from the feature map, skipping every other one along the rows and columns, to form new patches. These patches are then seamlessly combined into a single tensor and flattened out. Importantly, the temporal dimension remains untouched by this merging process, maintaining its original form in the output. As a result of halving the H and W dimensions, the channel dimension quadruples. To adjust this expansion, a 1 × 1 convolutional layer is employed, effectively doubling the initial channel dimension.

## Video swin-CLSTM transformer block

The major component of the overall architecture is the Video Swin-CLSTM Transformer block. As illustrated in Fig 2 (a), it consists of two consecutive units. The first unit adopts a structure similar to the Video Swin Transformer [13], comprising a 3D Window-based Multi-head Self-Attention (3D W-MSA) module followed by a Multilayer Perceptron (MLP) module [30]. The MLP module essentially consists of two Feedforward Network (FFN) layers with a Gaussian Error Linear Unit (GELU) activation function sandwiched in between, which performs nonlinear transformation and feature fusion after the self-attention operation. Layer Normalization (LN) [31] is applied prior to both the MSA and MLP modules, while residual connections are employed after each module, working synergistically to optimize gradient propagation throughout the model architecture. The second unit is generally similar to the first one, with the difference being the replacement of the

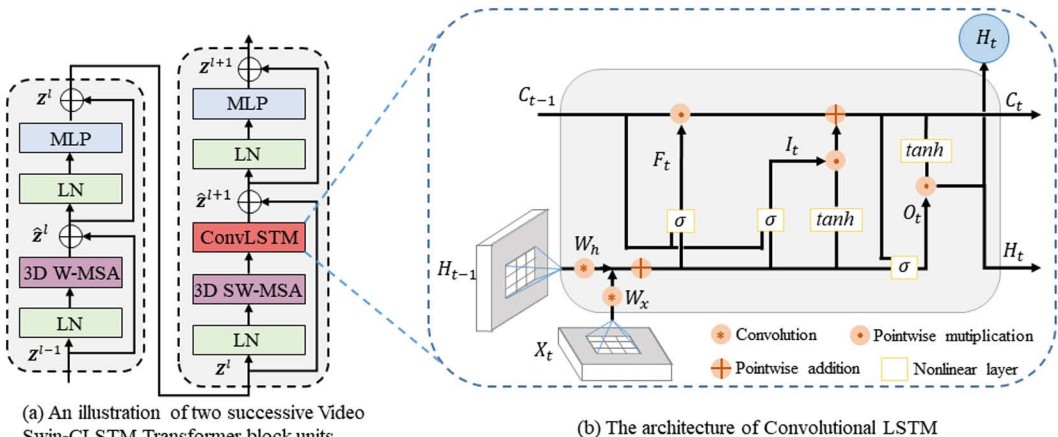

(a) An illustration of two successive Video Swin-CLSTM Transformer block units

(b) The architecture of Convolutional LSTM

**Fig 2. The details of Video Swin-CLSTM transformer block.** (a) Illustration of two successive Video Swin-CLSTM Transformer block units; **(b)** Overview of Convolutional LSTM.

3D W-MSA module with a 3D Shifted Window-based Multi-head Self-Attention (3D SW-MSA) module. Additionally, ConvLSTM networks are seamlessly integrated between the 3D SW-MSA and residual connection, utilizing the four-dimensional output tensor from 3D SW-MSA as its input. This integration ensures that the feature channel dimensions of ConvLSTM's input and output remain consistent, while maintaining the original positions of the MLP module unaltered. The key equations of the units' each portion are shown in Eq (1) below:

$$
\begin{aligned}
\hat{Z}^l &= 3DW-MSA(LN(Z^{l-1})) + Z^{l-1}, \\
Z^l &= MLP(LN(\hat{Z}^l)) + \hat{Z}^l, \\
\hat{Z}^{l+1} &= ConvLSTM(3DSW\text{-}MSA(LN(Z^l))) + Z^l, \\
Z^{l+1} &= MLP(LN(\hat{Z}^{l+1})) + \hat{Z}^{l+1}.
\end{aligned}
\tag{1}
$$

Here, $Z^{l-1}$ denotes the output features of the previous Video Swin Transformer block, $\hat{Z}^l$ denotes the output features of the 3D W-MSA module, $\hat{Z}^{l+1}$ denotes the output features of the ConvLSTM module, $Z^l$ and $Z^{l+1}$ denote the output features of the first and last unit MLP module respectively.

To improve the model's capacity for capturing temporal information, ConvLSTM [8] modules are incorporated into the architecture, as shown in Fig 2 (b). ConvLSTM extends LSTM [32] by integrating convolutional operations, enabling simultaneous processing of spatial and temporal information, particularly suitable for spatiotemporal sequence data. Specifically, the memory cell $C_t$ in ConvLSTM acts as a 3D information accumulator, accessed, updated, and reset by several self-parameterized controlling gates, with all information propagation occurring in 3D tensors. Upon receiving new input, if the input gate $I_t$ is activated, its information accumulates in the unit. Furthermore, if the forget gate $F_t$ is open, the previous cell state $C_{t-1}$ is "forgotten" during this process. Whether the latest cell output $C_t$ propagates to the final state $H_t$ is further controlled by the output gate $O_t$. A benefit of using memory cell and gates to regulate information flow is that gradients are trapped within the cell (also referred to as constant error carousels [33], preventing premature vanishing). The key equations of ConvLSTM are presented in Eq (2) below:

$$
\begin{aligned}
I_t &= \sigma(W_{xi} * X_t + W_{hi} * H_{t-1} + W_{ci} \circ C_{t-1} + b_i), \\
F_t &= \sigma(W_{xf} * X_t + W_{hf} * H_{t-1} + W_{cf} \circ C_{t-1} + b_f), \\
C_t &= F_t \circ C_{t-1} + I_t \circ \tanh(W_{xc} * X_t + W_{hc} * H_{t-1} + b_c), \\
O_t &= \sigma(W_{xo} * X_t + W_{ho} * H_{t-1} + W_{co} \circ C_t + b_o), \\
H_t &= O_t \circ \tanh(C_t).
\end{aligned}
\tag{2}
$$

Here, "$*$" denotes the convolution operator, "$\circ$" and "$+$" denote the pointwise multiplication and pointwise addition respectively. "$\sigma$" and "tanh" denote the activation function.

We have described the principle and formula of the calculation of the ConvLSTM module in Eq (2). Assume that the output of the video tensor after the global self-attention calculation of the 3D SW-MSA module is $X^l \in \mathbb{R}^{T \times H \times W \times C}$. When we use it as the input of the ConvLSTM, the expression Eq (3) for the combined calculation of the 3D SW-MSA module and the ConvLSTM module in Eq (1) is:

$$
\begin{aligned}
X^l &= 3DSW\text{-}MSA(LN(Z^l)), \\
H_t^l, C_t^l &= ConvLSTM(X_t^l, H_{t-1}^l, C_{t-1}^l), \forall t \in [1, T], \\
\hat{Z}^{l+1} &= Proj(H_T^l) + Z^l (\text{if} D \neq C)
\end{aligned}
\tag{3}
$$

Here, $t \in [1, T]$ iterates over temporal slices (video frames), $l \in [1, L]$ represents the layer index in a multi-layer architecture. $X^l \in \mathbb{R}^{T \times H \times W \times C}$ denotes the output spatiotemporal features with enhanced global dependencies, $X_t^l \in \mathbb{R}^{H \times W \times C}$ denotes the feature slice at time step t from $X^l$. $H_{t-1}^l$, $C_{t-1}^l \in \mathbb{R}^{H \times W \times D}$ denote the hidden state and cell state from the previous time

step respectively, their initial states are usually matrices or vectors filled with all zeros. $H_t^l$ represents the updated hidden state, encoding spatiotemporal context. Meanwhile, $H_T^l$ denotes the final hidden state after processing all T time steps, therefore, we only need to select $H_T^l$ as the output feature of the ConvLSTM module for subsequent calculations in the following modules. Particularly, here $\text{Proj}(\cdot)$ represents an optional 1×1 convolution operation for dimension alignment. When the hidden state dimension (D) of the ConvLSTM is equal to the number of feature channels (C), this operation is activated. Conversely, when $D \neq C$, $\text{Proj}(\cdot)$ acts as an identity mapping.

## Optical flow extraction and stacking

The optical flow information identifies the correspondence between the previous frame and the current frame by analyzing the changes in pixel values over time within an image sequence and the correlations between adjacent frames. This enables the calculation of motion information for objects between neighboring frames, effectively capturing instantaneous micro-motion details. Additionally, optical flow possesses appearance invariance, when an object moves in different viewpoints or backgrounds, the motion vectors between pixels in adjacent frames remain constant, thereby minimizing the impact of background variations. Consequently, combining optical flow information with spatial features enables a more detailed capture of the motion dynamics between frames. This advantage has also been verified in the subsequent ablation study shown in Table 2.

For the UCF-101 dataset, we perform preliminary data processing to prepare it for our experiments, which require the integration of optical flow information. Instead of training on entire videos, we focus on individual frames. First, we clean and organize the UCF-101 dataset, which consists of 101 video categories, each containing 25 groups of videos, with each group comprising 4–7 videos. As a result, each category includes approximately 100–175 videos. By utilizing functions from the OpenCV (Open Source Computer Vision Library), we extract RGB frames from the videos while maintaining the original resolution of 320×240 pixels. Due to the varying lengths of the videos, with an average duration of 7.21 seconds, the number of RGB frames per video also varies, averaging 184 frames per video. Each RGB frame is approximately 20 KB in size.

Subsequently, the TVL1 (Total Variation Regularization for L1 Optical Flow) algorithm [34] are applied to extract optical flow images from each video. Unlike RGB frames, optical flow captures the pixel-wise motion of objects over time by analyzing the changes in pixel values and the correlation between consecutive frames. For a video with 184 frames, we can extract 183 optical flow frames in one direction. To capture the motion information more comprehensively, we divide the optical flow images into two components: the x-direction and y-direction, representing the horizontal and vertical motion of each pixel, respectively. Consequently, the total number of optical flow frames extracted was double that of the frames in a single direction. To effectively fuse RGB frames with optical flow images, as shown in Fig 3, we stack the two types of optical flow data into a 2-channel image. We then add a third channel filled with zeros, creating a 3-channel optical flow image that encodes both motion direction and velocity information, making it compatible for model input.

## Sparse sampling and information fusion

To achieve efficient fusion while minimizing the model's computational load, we explore two different approaches. In the first approach, we alternate the insertion of optical flow images between RGB frames, as shown in Scheme 1 of Fig 4. Specifically, in the original continuous RGB sequence, pairs of adjacent RGB frames (e.g., Img_002, Img_003,…, Img_005, Img_006,..., etc.) are replaced with their corresponding fused three-channel optical flow images, while the remaining RGB frames (e.g., Img_001, Img_004, Img_007, Img_010,...) remain unchanged. Since the sampled RGB frames and optical flow images are evenly distributed frame-by-frame, optical flow data and RGB frames each account for half of the total input. Notably, the extracted optical flow images are generated based on the motion information between adjacent RGB frames. In other words, a single optical flow image encodes the semantic information of two adjacent RGB

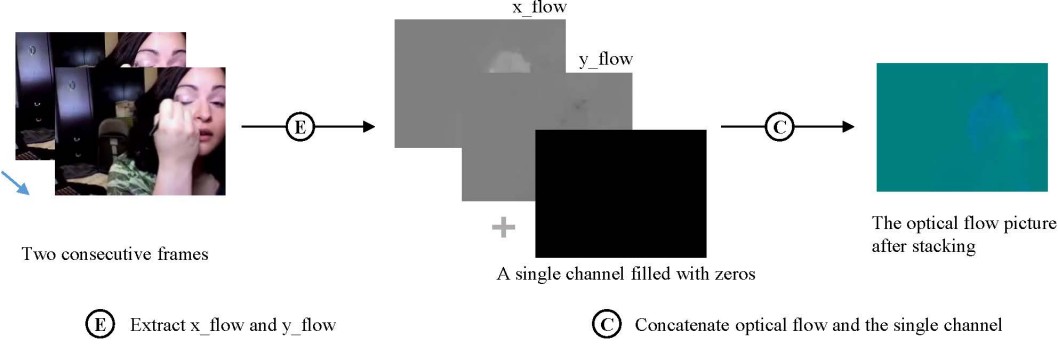

Fig 3. Optical flow extraction and stacking.

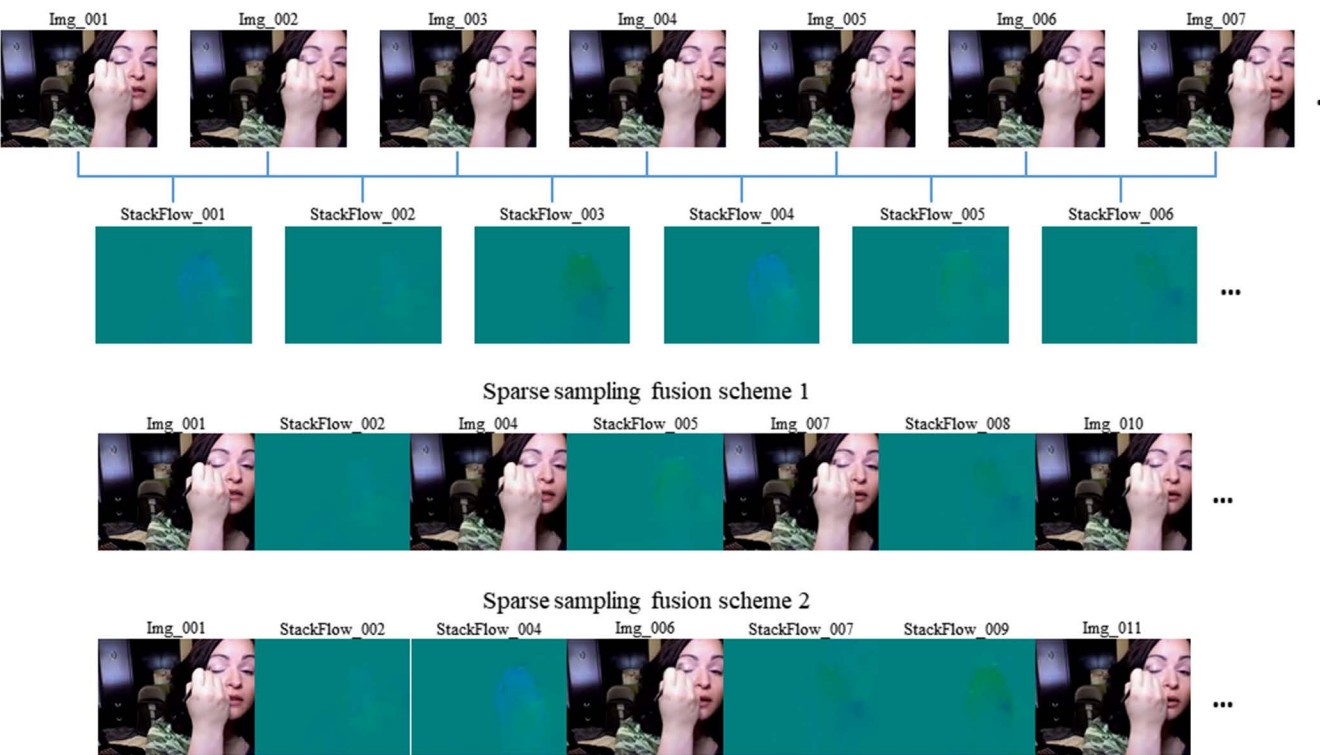

**Fig 4. Illustration of different sparse sampling fusion schemes.** In the top part of the figure, we describe the process of optical flow extraction, and then the specific implementation of sparse sampling fusion schemes 1 and 2 are shown in the bottom part of the figure, respectively.

frames, allowing us to express the same semantic content with fewer frames. Additionally, each synthetic optical flow image consists of grayscale channels, which carry significantly less information than RGB frames. As a result, the average size of an optical flow image is around 4 KB, compared to an average of 20 KB for an RGB frame, thereby reducing hardware storage requirements. Experimental results using this approach (The input is RGB+Flow) demonstrated improved accuracy compared to training solely on RGB frames, while also reducing computational costs, as shown in Table 1. These findings suggest that incorporating optical flow data into the Video Swin Transformer model enhances its ability to capture human action information.

**Table 1. Comparison to Swin Transfomer-Based algorithms on _UCF-101_. Top-1 and Top-5 values represent the mean and standard deviation (Mean±Std), calculated from five repetitions under identical configurations. "RGB+Flow" (50%) and "LessRGB+Flow" (67%) denote optical flow ratios. The magnitudes are Giga (10⁹) and Mega (10⁶) for FLOPs and Param (Parameters) respectively. Inference (ms/frame) and memory (GB) measure speed and memory usage.**

| Method | Input | Pretrain | Datasets | Top-1 (Mean±Std) | Top-5 (Mean±Std) | FLOPs (G) | Param (M) | Inference (ms/frame) | Memory (GB) |
|---|---|---|---|---|---|---|---|---|---|
| Swin-T | RGB | – | ImageNet-1K | 81.2±0.05 | 95.5±0.03 | 4.5 | – | 4.7 | 4.56 |
| Swin-LSTM (ours) | RGB | – | ImageNet-1K | **81.48**±0.1 | **95.6**±0.05 | 6.6 | – | 4.9 | 4.67 |
| Video Swin-T | RGB | ImageNet-1K | UCF-101 | 89.6±0.15 | 97.4±0.06 | 93 | 29.8 | 9.1 | 10.2 |
| | RGB+Flow | ImageNet-1K | UCF-101 | **92.2**±0.1 | **99.1**±0.08 | 91 | 28.9 | 9.0 | 9.9 |
| | LessRGB+Flow | ImageNet-1K | UCF-101 | 91.7±0.2 | 98.6±0.08 | 76 | 28.7 | 8.5 | 9.3 |
| Video Swin-CLSTM(ours) | RGB | ImageNet-1K-LSTM | UCF-101 | 91.5±0.1 | 98.7±0.07 | 273 | 99.2 | 14.8 | 17.9 |
| | RGB+Flow | ImageNet-1K-LSTM | UCF-101 | **92.8**±0.1 | **99.4**±0.1 | 261 | 98.1 | 14.4 | 17.2 |
| | LessRGB+Flow | ImageNet-1K-LSTM | UCF-101 | 92.5±0.15 | 99.3±0.09 | 239 | 98.0 | 13.7 | 16.4 |

To further investigate the role of optical flow information in human action recognition, we conduct a second set of experiments. Inspired by the notion that appearance changes slowly while motion changes rapidly [35], we adopt a sparser sampling strategy and double the number of RGB frames replaced in Scheme 1 (using groups of four adjacent frames), as depicted in Scheme 2 of Fig 4. Specifically, in the original continuous RGB sequence, four adjacent RGB frames (e.g., Img_002, Img_003, Img_004, Img_005,..., Img_007, Img_008, Img_009, Img_010,...) are replaced with their corresponding two fused optical flow images, while the remaining RGB frames (e.g., Img_001, Img_006, Img_011,...) remain unchanged. As a result, optical flow data accounts for 67% of the total input, and these images (referred to as "LessRGB+Flow") are used as tokens fed into the model. The experimental results based on our model in Table 1 of the Experiments section showed that the Top-1 accuracy of Scheme 2 decreased by 0.3% compared to Scheme 1, but the computational cost was reduced by 22 GFLOPs. Compared to the original model (Input: RGB), Scheme 2 achieved a 1% improvement in Top-1 accuracy and reduced the computational cost by 34 GFLOPs. These findings indicate that similar accuracy can be obtained with lower computational overhead.

## Experiments

### Datasets

The UCF-101 dataset [36] is a widely used benchmark in HAR, released by the Center for Research in Computer Vision (CRCV) at the University of Central Florida in 2012. It consists of 13,320 video clips sourced from YouTube, categorized into 101 distinct action classes. Each class is divided into 25 groups, with each group containing 4–7 video samples. The action classes are grouped into five main categories: body movements, human-human interactions, human-object interactions, playing musical instruments, and sports activities. The video clips vary in length, with an average duration of approximately 7.21 seconds, and are captured at a resolution of 320×240 pixels and a frame rate of 25 FPS. The dataset poses significant challenges due to its diverse recording conditions, including variations in lighting, camera motion, partial occlusions, and the presence of low-quality frames. It can be accessed at https://www.crcv.ucf.edu/research/data-sets/ucf101/.

UCF-101 contains videos with a wide range of backgrounds and viewpoints, where actions can be obscured by complex backgrounds or shifts in perspective. For instance, in action classes such as _Basketball Shooting_, the action may occur in different environments with varying crowd backgrounds, complicating recognition. Additionally, certain action classes, such as _Typing_ or _Knitting_, involve subtle hand movements that entail minimal spatial displacement, making them

crucial for accurate recognition. The variability in video duration, with an average length of 7.21 seconds, requires models capable of handling inputs of varying lengths while effectively capturing long-term temporal dependencies. These characteristics make UCF-101 a challenging yet valuable dataset for evaluating and advancing action recognition algorithms.

## Experiments setup

We use Video Swin-Tiny (Video Swin-T) [13] as the basic architecture for the experiments. For the pre-training model of the experiments, instead of using the traditional ImageNet-1k pre-training model based on Swin Transfomer, we train from scratch on ImageNet-1k based on the 2D Swin-LSTM model architecture newly proposed in this paper. Similarly, we add the LSTM module with layer number 1 and the rest of the parameters as default, inside the 2nd and 4th stages of Swin Transfomer. The parameter settings for training the pre-trained model ImageNet-1k-CLSTM of the model Swin-CLSTM mainly follow those of Swin-T [15]. We also employ an AdamW optimizer for 300 epochs using a cosine decay learning rate scheduler and 20 epochs of linear warm-up. A batch size of 1024, an initial learning rate of 0.001, a weight decay of 0.05 and the Trivial Augment data enhancement method are used. For Video Swin-CLSTM, we embed ConvLSTM with a convolution kernel size of 3×3, the dimension of the hidden layer state equal to the dimension of the input features and the number of layers is 1 in the SwinTransformerBlock3D module of Stage 2 and Stage 4, to emphasize, ConvLSTM is located between 3D W-MSA/3D SW-MSA and residual connection. Four NVIDIA Quadro RTX 6000 GPUs, each with 24GB of video memory, are utilized. For the alternative configurations of Video Swin-CLSTM, we utilize an AdamW optimizer over 50 epochs (vs. Video Swin-T's 30 epochs) to more comprehensively optimize the model's parameters, integrating a cosine decay learning rate scheduler with a 5-epoch linear warm-up phase (vs. 2.5 epochs in baseline) for smoother transition given the training schedule. A batch size of 16 is adopted (vs. Video Swin-T's 64) due to lower GPU memory compared to the baseline model experiments and the additional computational overhead brought by the ConvLSTM module, and we further scale down the backbone learning rate by a factor of 0.1 (vs. full 0.001 in baseline) to preserve pre-trained feature stability while allowing new ConvLSTM layers to learn aggressively. For input, we sample an average of 128 frames (with optical flow images extracted from them) from each full-length video, employing a temporal stride of 1 (vs. baseline's stride = 2) to capture finer motion patterns, with spatial resolution maintained at 224×224 as in Video Swin-T, yielding 128×56×56 input 3D tokens. Aligning with Video Swin-T, we apply a stochastic depth rate of 0.3 (matching baseline) and weight decay of 0.05 (identical to baseline). For inference, we adhere to the methodology outlined in [27], utilizing 4×3 views, we uniformly sample 4 clips in the temporal dimension from each video, and for each clip, we resize the shorter spatial side to 224 pixels before taking 3 crops of 224×224 pixels that encompass the longer spatial axis. The final score is derived as the mean of the scores across all views.

## Training and testing

For training, we use a dataset of 9,537 videos, with approximately 40% (3,783 videos) randomly selected for validation and testing. Instead of directly using the ImageNet-1k pre-trained model based on the Swin Transformer, we modify the Swin Transformer architecture to better align with our Video Swin-CLSTM model. Specifically, we embed the 2D version of the LSTM into the second and fourth stages of the Swin Transformer. Subsequently, we train this modified model from scratch using the ImageNet-1k dataset to minimize the influence of the original pre-trained model on the structure of the Video Swin-CLSTM. After obtaining the new pre-trained model, we use it to initialize the weights of the Video Swin-CLSTM model, which is subsequently trained on the UCF-101 dataset. During training, we apply the auto-scaling learning rate strategy, which automatically adjusts the learning rate based on the model's performance on the validation set. This approach accelerates training and enhances the model's generalization capability.

## Results

In action recognition tasks, model performance is quantified by Top-1 and Top-5 accuracy (%). Top-1 accuracy measures the proportion of test samples where the model's highest-confidence prediction matches the ground-truth label,

while Top-5 accuracy evaluates whether the true label is within the top five predictions, which is crucial for distinguishing ambiguous or fine-grained actions. To ensure rigorous comparisons, all experiments were repeated 5 times under identical configurations (dataset splits, hyperparameters, hardware), with randomness controlled through distinct random seeds (0–4) governing parameter initialization, data augmentation, and batch ordering. This multi-run protocol isolates the impact of stochastic factors, ensuring reported results reflect consistent model capability.

Table 1 compares our proposed model with various Swin Transformer-Based algorithms. With the exception of the first group, all other groups employ pre-trained models. For first group, we evaluated two approaches: The Swin-T model without the LSTM module and the Swin-LSTM model with the LSTM module. Despite the additional computational overhead (+2.1 GFLOPs, 4.5→6.6) introduced by the LSTM module, the Swin-LSTM model achieved 0.28% improvement in Top-1 accuracy (81.48%±0.1% vs 81.2%±0.05%), with a narrow standard deviation indicating high stability. This result is due to LSTMs' enhances ability to capture temporal dependencies and long-term contexts for 2D image, and demonstrates that the LSTM module can enhances the accuracy of Transformer backbone models, which provides strong support for the subsequent proposal of our new model.

In the second group of experiments, pre-trained models based on ImageNet-1k were utilized. The Video Swin-T model incorporating optical flow information exhibited superior overall accuracy compared to the Video Swin-T model without optical flow. Specifically, the Video Swin-T model with RGB+Flow input achieved an accuracy of 92.2%, surpassing the RGB-only model (89.6%) by 2.6% in Top-1 mean accuracy while incurring less computational cost (↓2 GFLOPs) and peak memory usage (↓0.3 GB). The Video Swin-T model with LessRGB+Flow input achieved a mean accuracy of 91.7%, which is 2.1% higher than the RGB-only model but 0.5% lower than the RGB+Flow model. However, it resulted in a reduction of 17 GFLOPs and 0.6ms in computational cost and inference time, compared to the RGB+Flow model's reduction of 2 GFLOPs and 0.1ms. The reason for these results is that optical flow captures motion dynamics and emphasises important changes while reducing redundant information. In addition, fusing of RGB data and optical flow reduces the computational requirements for complete RGB frames and improves computational efficiency. This also indicates that optical flow effectively conveys rich motion semantics with less data.

The third group of experiments highlights the impact of the ConvLSTM module on model performance. Using our pre-trained model, ImageNet-1K-LSTM, we observed higher overall mean accuracy compared to the second group, which did not include the ConvLSTM module. The RGB-input model augmented with a ConvLSTM module achieved a mean accuracy of 91.5%, a 1.9% improvement over the model without ConvLSTM (89.6%). However, this gain came with increased computational overhead and slower inference times (from 9.1ms to 14.8ms). For the other two configurations "RGB+Flow" and "LessRGB+Flow", the inclusion of ConvLSTM modules improved mean Top-1 accuracy by 0.6% (92.2% to 92.8%) and 0.8% (91.7% to 92.5%), respectively, again at the cost of higher computational overhead and slower inference. These trade-offs are reasonable and can be explained by the additional processing introduced by the ConvLSTM module. Similar to the second group, the model with RGB+Flow input attained a highest mean accuracy of 92.8%, which is 1.3% higher than the RGB-only input model (91.5%). The Video Swin-CLSTM model with LessRGB+Flow input achieved a mean accuracy of 92.5% and demonstrated the greatest reduction in computational cost and inference time, decreasing by 34 GFLOPs (273→239) and 1.1ms (14.8→13.7) compared to the RGB input Video Swin-CLSTM model. Similar to the LSTM in the first group experiment, this study introduces ConvLSTM to adapt to three-dimensional video data. ConvLSTM not only significantly enhances the Transformer-Based video backbone's ability to capture long-term semantic information but also effectively addresses issues related to model robustness caused by speed, viewpoint, and background variations by leveraging optical flow information, thereby improving the overall performance of the model.

## Ablation study

### Sparse sampling fusion scheme

In the section Sparse Sampling and Information Fusion, we investigated two sparse sampling fusion schemes, with Scheme 1 serving as the default configuration for the model. To further assess the impact of optical flow information

on the experimental outcomes, six groups of experiments were performed. In the first group, the proportion of optical flow information was set to 0, meaning that only RGB frames were used as model input. In the second group, the input sequence was selected as (Img_001, Image_002, StackFlow_003, Img_005, Image_006, StackFlow_007,..., etc.), resulting in an optical flow proportion of 33%. The third and fourth groups adopted the same sparse sampling fusion schemes as those in the section Sparse Sampling and Information Fusion (Scheme 1 and Scheme 2), with optical flow proportions of 50% and 67%, respectively (see Fig 4 for details). In the fifth group, based on Scheme 2 from Fig 4, the stride was expanded to increase the sparsity of the sampling, which involved removing additional intermediate RGB frames. Specifically, the proportion of optical flow images in the input sequence was adjusted from 67% to 80%, modifying the sequence to (Img_001, StackFlow_002, StackFlow_004, StackFlow_006, StackFlow_008, Img_009,..., etc.). In the sixth group, only the optical flow images numbered with odd numbers were used as input (e.g., StackFlow_001, StackFlow_003, Stack-Flow_005,..., etc.), resulting in an optical flow proportion of 100%. Table 2 summarizes the results of these experiments, with top-1/5 accuracy presented as mean values derived from 5 repetitions under identical configurations with controlled randomness via seeds. The same definition applies to the results in subsequent ablation experiments.

The findings indicate that a 50% proportion of input optical flow images yields the optimal performance, it shows Top-1 and Top-5 accuracies of 92.8% and 99.4% respectively. This reflects improvements of +1.3% in Top-1 and +0.7% in Top-5 accuracy compared to the first experiment without optical flow information. Notably, models incorporating optical flow information consistently outperformed those without it, except in the sixth experiment. This enhancement is attributed to the optical flow's capability to capture motion information and improve temporal sequence modeling, while also reducing computational cost and inference time. Although continuing to increase the percentage of input optical flow images (>50%) leads to a slight decrease in Top-1 and Top-5 accuracy, the reduction in FLOPs and inference time is more substantial, as shown in Fig 5 (b). In the final experiment, where only optical flow images were used, the model's accuracy was the lowest, likely due to the lack of static appearance information provided by RGB frames. When RGB frame information is reduced below a certain threshold, the model lacks sufficient visual context, resulting in decreased recognition accuracy. Thus, balancing these two types of information is essential for optimal model performance.

### Robustness of optical flow

In the section Sparse Sampling Fusion Scheme, we explored the impact of the proportion of optical flow fusion on the performance of the model, determined the optimal proportion of optical flow fusion, and reached the conclusion that optical flow can improve the accuracy of the model to a certain extent. However, we did not provide a detailed analysis of how optical flow impacts performance in different motion scenarios (e.g., fast motions, occlusive motions, subtle motions). To this end, we separately conducted experiments on improving the model accuracy on different sub-datasets of UCF-101 with complex scenarios to further explore the anti-interference ability and robustness of optical flow. We construct three motion-specific subsets from the UCF-101 dataset through quantitative analysis. The Fast Motion subset includes 23

**Table 2. Ablation study on the percentage of input optical flow image with Video Swin-CLSTM Transformer on *UCF-101*.**

| Percentage of input optical flow image | Top-1 (%) | Top-5 (%) | FLOPs (G) | Inference (ms/frame) | Memory (GB) |
|---|---|---|---|---|---|
| 0 | 91.5 | 98.7 | 273 | 14.8 | 17.9 |
| 33% | 92.3 | 99.3 | 269 | 14.6 | 17.6 |
| 50% | **92.8** | **99.4** | 261 | 14.4 | 17.2 |
| 67% | 92.5 | 99.3 | 239 | 13.7 | 16.4 |
| 80% | 91.8 | 98.9 | 218 | 12.9 | 15.1 |
| 100% | 89.7 | 97.6 | 187 | 11.8 | 13.6 |

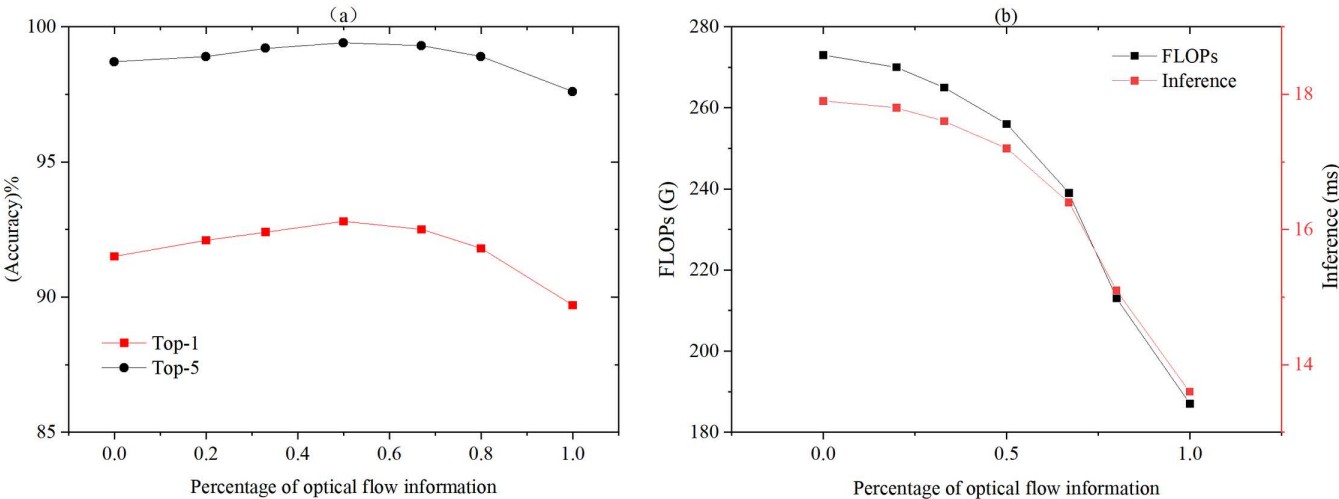

**Fig 5. Effect of the proportion of optical flow input on model accuracy (a) and computational overhead (b).**

classes (22.8% of the total, e.g., *BasketballShooting*, *TennisSwing*), where over 75% of video frames have optical flow magnitudes exceeding 15 pixels/frame, indicating significant displacement. The occlusion subset consists of 17 classes (16.8% of the total, e.g., *BoxingPunchingBag*, *SoccerJuggling*), selected based on manual annotations showing over 30% frame-level occlusion coverage due to persistent inter-object interactions. The Subtle Motion subset contains 19 classes (18.8% of the total, e.g., *ApplyEyeMakeup*, *Typing*), characterized by localized movements with average flow magnitudes below 5 pixels/frame. After that, we conducted experiments on the baseline and Video Swin-CLSTM respectively, and used Flow Gain to measure the role of the contribution of optical flow information. In order to better adapt to the characteristics of the data, the number of frames we sampled varies in different scenarios (e.g.,160 frames for fast motions, 96 frames for subtle motions). For consistency, we use the optimal optical flow ratio of 50% as the default configuration for RGB+Flow in our experiments. In Video Swin-CLSTM, the ConvLSTM modules are by default placed in Stage2 and Stage4, positioned between the 3D-SWMSA modules and the residual connections. (including subsequent ablation studies, unless otherwise specified). The results are shown in Table 3.

**Table 3. An ablation study of the robustness of the optical flow under different motion scenarios.**

| Scenarios | Method | Input | Top-1 (%) | FLOPs (G) | Flow Gain (△%) |
|---|---|---|---|---|---|
| Fast motions | Video Swin-T | RGB only | 88.8 | 114 | – |
| | | RGB+Flow | 89.1 | 110 | **+0.3** |
| | Video Swin-CLSTM | RGB only | 91.2 | 339 | – |
| | | RGB+Flow | 91.3 | 323 | **+0.1** |
| Occlusive motions | Video Swin-T | RGB only | 79.2 | 93 | – |
| | | RGB+Flow | 82.6 | 91 | **+3.4** |
| | Video Swin-CLSTM | RGB only | 84.3 | 273 | – |
| | | RGB+Flow | 86.4 | 261 | **+2.1** |
| Subtle motions | Video Swin-T | RGB only | 81.4 | 68 | – |
| | | RGB+Flow | 84.5 | 66 | **+3.1** |
| | Video Swin-CLSTM | RGB only | 87.8 | 201 | – |
| | | RGB+Flow | 89.3 | 192 | **+1.5** |

Table 3 highlights the importance of optical flow in improving action recognition robustness in challenging scenarios. For fast motions, where the action amplitude is more pronounced and easier for the model to capture and learn, the overall accuracy is highest. However, this also reduces the model's discriminative capacity, resulting in a modest accuracy improvement (Swin-T by 0.3% and CLSTM by 0.1%). In occlusion scenarios, accuracy drops significantly (by approximately 7.2%) due to target obstruction. However, optical flow compensates for the loss of spatial information, while the ConvLSTM module maintains motion context during occlusions. This synergy improves the temporal consistency, raising the accuracy of Video Swin-T and Video Swin-CLSTM by 3.4% (79.2%→82.6%) and 2.1% (84.3%→86.4%), respectively. For subtle motions, combining optical flow with RGB inputs increases the Top-1 accuracy of Video Swin-T by 3.1% (81.4%→84.5%) and Video Swin-CLSTM by 1.5% (87.8%→89.3%), as optical flow encodes motion trajectories, enriching the feature representation. Across all tested scenarios, models with optical flow consistently outperform those without it, with reduced computational costs. This performance boost is attributed to the appearance invariance and compact information encoding of optical flow. Furthermore, the enhanced temporal coherence from optical flow, combined with the Video Swin-CLSTM architecture, creates a robust framework for more efficient action recognition. To better illustrate the robustness of optical flow against interference, motion patterns under different scenarios are visualized in Fig 6.

## Position of the ConvLSTM Module

We examined embedding the ConvLSTM module at various positions within the second unit of the Video Swin-CLSTM Transformer block. These positions included before the 3D SW-MSA module, between the 3D SW-MSA module and the residual connection, and after the residual connection. Units with the 3D SW-MSA module involve a sliding window operation for information interaction, and incorporating the ConvLSTM module in these units can capture richer long-term temporal information. However, adding redundant ConvLSTM modules does not enhance model performance but significantly increases parameter count, computational complexity, and memory usage, which reduces the model's interpretability. Therefore, we only explored adding the ConvLSTM module in the second unit of the Video Swin-CLSTM Transformer block, without considering its inclusion in the first unit. For consistency, we embedded the ConvLSTM module in all stages (not necessarily the optimal strategy). The results are summarized in Table 4.

The experimental results indicate that embedding the ConvLSTM module between the 3D SW-MSA module and the residual connection yields the best performance. This configuration achieved a Top-1 accuracy of 84.7%, an increase of 2.8% compared to placing the ConvLSTM module before and a 0.6% increase compared to placing it after the residual connection. Embedding the ConvLSTM module in every stage required additional computational overhead, resulting in a computational cost 98.1 GFLOPs higher than the configuration in Table 1 (261 GFLOPs). Although the computational

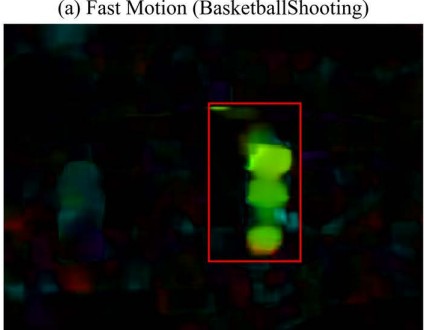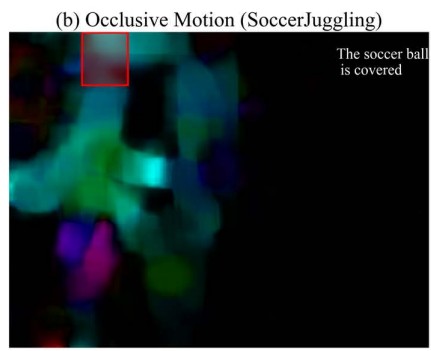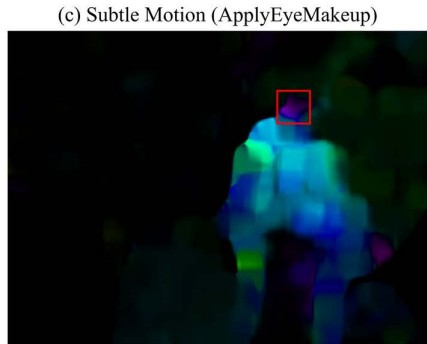

(a) Fast Motion (BasketballShooting)  (b) Occlusive Motion (SoccerJuggling)  (c) Subtle Motion (ApplyEyeMakeup)

**Fig 6. Visualization of optical flow under different motion scenarios.** (a) Fast Motion (*BasketballShooting*); (b) Occlusive Motion (*SoccerJuggling*); (c) Subtle Motion (*ApplyEyeMakeup*).

**Table 4. An ablation study of the position of the ConvLSTM module within the second unit of each stage's Video Swin-CLSTM Transformer block using the *UCF-101* dataset.**

| The position of the ConvLSTM module | Top-1 (%) | Top-5 (%) | FLOPs (G) | Gradient Var (↓) |
|---|---|---|---|---|
| Before the 3D SW-MSA module. | 82.5 | 94.7 | 357.3 | 0.38 |
| Between the 3D SW-MSA module and residual connection. | **85.3** | **96.4** | 359.1 | 0.11 |
| After the residual connection. | 84.7 | 96.2 | 358 | 0.29 |

costs and parameter counts were similar across the three configurations, adding the ConvLSTM module after the 3D SW-MSA module, which performs sliding window operations, allows it to access globally interacted semantic information directly, enhancing its ability to capture long-term dependencies. In contrast, inserting the ConvLSTM module before the 3D SW-MSA module distances it from global semantic information, reducing performance. When placed after the residual connection, the model may also experience instability due to the fusion of complex features from different layers.

To better explain this phenomenon, we conducted a theoretical analysis of ConvLSTM positioning, focusing on two key aspects: information flow analysis and gradient propagation stability. For information flow analysis, the optimal placement between the 3D SW-MSA module and the residual connection establishes a dual-path fusion mechanism:

$$X_{\text{out}} = \underbrace{\text{ConvLSTM}(3\text{DSW} - MSA(X_{in}))}_{Global-temporal path} + \underbrace{X_{in}}_{Local-spatial path}$$

(4)

The 3D SW-MSA first captures global dependencies across spatiotemporal windows (receptive field: 128 frames), generating semantically enriched features. The subsequent ConvLSTM then models temporal dynamics within this global context, while the residual path directly propagates low-level motion cues. This hierarchical design ensures decoupled learning of spatial-semantic and temporal-relational features, avoiding interference between global attention and local motion patterns.

For gradient propagation stability, we introduce a gradient variance metric to quantify optimization robustness. The variance is computed as:

$$\text{Var}(G) = \frac{1}{N} \sum_{i=1}^{N} (\|\nabla_{\mathsf{x}_i}\mathcal{L}\|_2 - \mu_G)$$

(5)

where $\nabla_{\mathsf{x}_i}\mathcal{L}$ denotes the gradient of the loss $\mathcal{L}$ with respect to input $\mathsf{x}_i$, and $\mu_G = \frac{1}{N} \sum_{i=1}^{N} \|\nabla_{\mathsf{x}_i}\mathcal{L}\|_2$ represents the mean gradient norm across N samples. As shown in Table 4, the gradient variance of the optimal position was reduced by 71% and 63%, respectively, compared to the pre-SW-MSA placement (0.11 vs. 0.38) and the post-residual placement (0.11 vs. 0.29), indicating significantly stabilized backpropagation. The residual connection acts as a gradient highway to mitigate vanishing/exploding gradients, while the ConvLSTM benefits from stable gradients propagated through SW-MSA's globally contextualized features. Theoretical principles of differential feature decoupling and error propagation stability jointly justify this design's superiority.

## Distribution of the video swin-CLSTM block

We determined the optimal position for the ConvLSTM module within the Video Swin-CLSTM Transformer block through ablation studies. However, we did not investigate the optimal distribution of the Video Swin-CLSTM Transformer block (Video Swin-CLSTM block) across different stages. As shown in Tables 1 and 4, embedding the ConvLSTM module in

every stage results in a maximum Top-1 accuracy of 85.3%, while embedding it only in the 2nd and 4th stages achieves a Top-1 accuracy of 92.8%, a difference of 7.5%. This demonstrates that the distribution of the Video Swin-CLSTM block significantly impacts model performance.

Previous experiments indicate that excessive redundancy or simplification of the Video Swin-CLSTM block reduces model effectiveness. Consequently, we consider only 2-stage or 3-stage distributions for the Video Swin-CLSTM block. For a 2-stage distribution, the possible configurations are (Stage1, Stage2), (Stage1, Stage3), (Stage1, Stage4), (Stage2, Stage3), (Stage2, Stage4), and (Stage3, Stage4), resulting in 6 patterns. For a 3-stage distribution, the configurations are (Stage1, Stage2, Stage3), (Stage1, Stage2, Stage4), (Stage1, Stage3, Stage4), and (Stage2, Stage3, Stage4), resulting in 4 patterns. However, since Stage 3 of our model architecture (Fig 1) includes six Video Swin-CLSTM block units, embedding the block in this stage would increase computational overhead and reduce interpretability. Thus, for the 3-stage distribution, we consider only (Stage1, Stage2, Stage4) as the final configuration. The final experimental results are presented in Table 5.

The experimental results show that distributing the ConvLSTM modules across Stage 2 and Stage 4 achieves the highest Top-1 accuracy of 92.8% and Top-5 accuracy of 99.4%, with a computational cost of 261 GFLOPs and a parameter count of 98.1 million. This distribution effectively captures long-term dependencies in video data while maintaining computational efficiency. In contrast, distributing the ConvLSTM modules across all stages, while capturing more feature information, significantly increases computational cost and parameter count (359.1 GFLOPs and 132.1 million, respectively), resulting in a lower Top-1 accuracy of 85.3%. This suggests that redundant ConvLSTM modules not only fail to improve model performance but also introduce additional computational burden and negative effects.

Other distribution strategies, such as placing ConvLSTM modules in Stage 1 and Stage 4 or in Stage 1 and Stage 2, achieved Top-1 accuracies of 91.9% and 90.8%, respectively. While these configurations demonstrated good performance, they still fell short of the Stage 2 and Stage 4 distribution. Specifically, the Stage 1 and Stage 4 configuration, despite its near-optimal performance, had a slightly higher computational cost and parameter count, at 272 GFLOPs and 101 million, respectively.

Furthermore, distributing ConvLSTM modules across three stages was less effective than the two-stage distribution. For example, the configuration with modules in Stage 1, Stage 2, and Stage 4 achieved a Top-1 accuracy of 86.6% and a Top-5 accuracy of 96.8%, with a computational cost of 320 GFLOPs and a parameter count of 123.5 million. Similarly, the configuration with modules in Stage 1, Stage 2, and Stage 3 resulted in the same Top-1 and Top-5 accuracies, with identical computational cost and parameter count. These results indicate that embedding the ConvLSTM module across multiple stages may lead to an overly complex model structure, generating excessive feature information and causing overfitting. Consequently, it has a negative impact on the stability and performance of the model.

**Table 5. An ablation study of Video Swin-CLSTM Transformer block with different distribution in each stage on *UCF-101*. ConvLSTM embedding position is between the 3D SW-MSA module and the residual connection, "S" denotes "Stage".**

| The distribution stages of Video Swin-CLSTM Transformer block | Top-1 (%) | Top-5 (%) | FLOP (G) | Param (M) |
|---|---|---|---|---|
| (S1, S2) | 90.8 | 98.1 | 284 | 107.6 |
| (S1, S3) | 86.8 | 96.9 | 324 | 125 |
| (S1, S4) | 91.9 | 99.1 | 272 | 101 |
| (S2, S3) | 87.6 | 97.3 | 319 | 124.3 |
| (S2, S4) | **92.8** | **99.4** | 261 | 98.1 |
| (S3, S4) | 87.9 | 97.8 | 314 | 121 |
| (S1, S2, S4) | 86.6 | 96.8 | 320 | 123.5 |
| (S1, S2, S3, S4) | 85.3 | 96.4 | 359.1 | 132.1 |

## Investigation of multi-layer convLSTM stacking

In the sections Position of the ConvLSTM Module and Sec Distribution of the Video Swin-CLSTM Block, we have already explored the optimal embedding position of the ConvLSTM module and the best distribution of the Video Swin-CLSTM block. To eliminate other potential instability factors, such as the impact of the number of ConvLSTM layers on model performance, we conducted an ablation study. Based on the results from the previous ablation experiments, redundant distribution does not improve model performance. Therefore, we only considered two-stage and three-stage distributions of the Video Swin-CLSTM block, with the ConvLSTM module placed at the optimal embedding position, and num_layer parameter controls the number of stacked ConvLSTM layers.

The experimental results in Table 6 and Fig 7 demonstrate that stacking ConvLSTM layers consistently degrades model accuracy and efficiency. For example, increasing layers from 1 to 3 in Stage2＋4 reduces Top-1 accuracy by 4.3% (92.8%→88.5%) and Top-5 by 0.7% (99.4%→98.7%), while computational costs surge by 24.5% in FLOPs (261→325) and 17% in latency (14.4ms→16.9ms). Extending ConvLSTM to early stages (Stage1＋2＋4) exacerbates these issues, causing a 4.5% Top-1 accuracy drop (86.6%→82.1%) and 38.1% higher FLOPs (320→442). The optimal single-layer ConvLSTM in Stage2＋4 achieves both peak accuracy (92.8% Top-1) and efficiency (261 FLOPs, 14.4ms latency). These phenomena stem from three limitations: deep layers oversmooth temporal features, erasing high-frequency motion details essential for fine-grained actions; low-level stages (e.g., Stage1) face spatial-temporal conflicts as they prioritize static semantics over dynamic patterns; and optimization instability arises from vanishing gradients and redundant parameters that disrupt temporal signal propagation. The findings highlight the superiority of shallow, strategically placed temporal modeling over rigid layer stacking. Future work may prioritize adaptive temporal depth over brute-force stacking.

## Evaluation of pre-trained models

In the Results section, the experimental results demonstrated that our custom pre-trained model, ImageNet-1K-LSTM, achieved optimal performance. However, we had not previously explored the impact of using the original pre-trained model, ImageNet-1K, from the Video Swin Transformer on experimental outcomes. To address this, we conducted an ablation study comparing these two pre-trained models, with results detailed in the Table 7.

The results indicate that the ImageNet-1K-LSTM pre-trained model achieved the highest Top-1 accuracy of 92.8% on the UCF-101 dataset, while the original ImageNet-1K pre-trained model achieved a maximum accuracy of 90.8%. This suggests that our custom pre-trained model excels in processing video data and capturing long-term dependencies. Specifically, this training approach encodes spatiotemporal modeling priors into the initialization parameters through a structurally matched pre-training model, enabling more efficient transfer learning in video tasks. This strategy aligns with the core principle of the "pre-training–fine-tuning" paradigm in deep learning, where the closer the pre-training domain and structure are to the target task, the better the transfer performance. Furthermore, the fusion of RGB and optical flow inputs was found to outperform the use of RGB inputs alone. With the ImageNet-1K-LSTM pre-trained model, the Top-1

**Table 6. Impact of convLSTM layer stacking.**

| Stage Distribution | num_layer | Total_Num | Top-1 (%) | Top-5 (%) | FLOPs (G) | Param (M) | Inference (ms/frame) | Memory (GB) |
|---|---|---|---|---|---|---|---|---|
| Stage2＋4 | 1 | 4 | **92.8** | **99.4** | 261 | 98.1 | 14.4 | 17.2 |
| | 2 | 8 | 91.1 | 99.1 | 292 | 112.3 | 15.6 | 18.6 |
| | 3 | 12 | 88.5 | 98.7 | 325 | 128.5 | 16.9 | 20.1 |
| Stage1＋2＋4 | 1 | 6 | **86.6** | **96.8** | 320 | 123.5 | 16.5 | 19.9 |
| | 2 | 12 | 84.2 | 96.1 | 383 | 161.0 | 19.2 | 22.6 |
| | 3 | 18 | 82.1 | 95.2 | 442 | 199.2 | 21.8 | 23.9 |

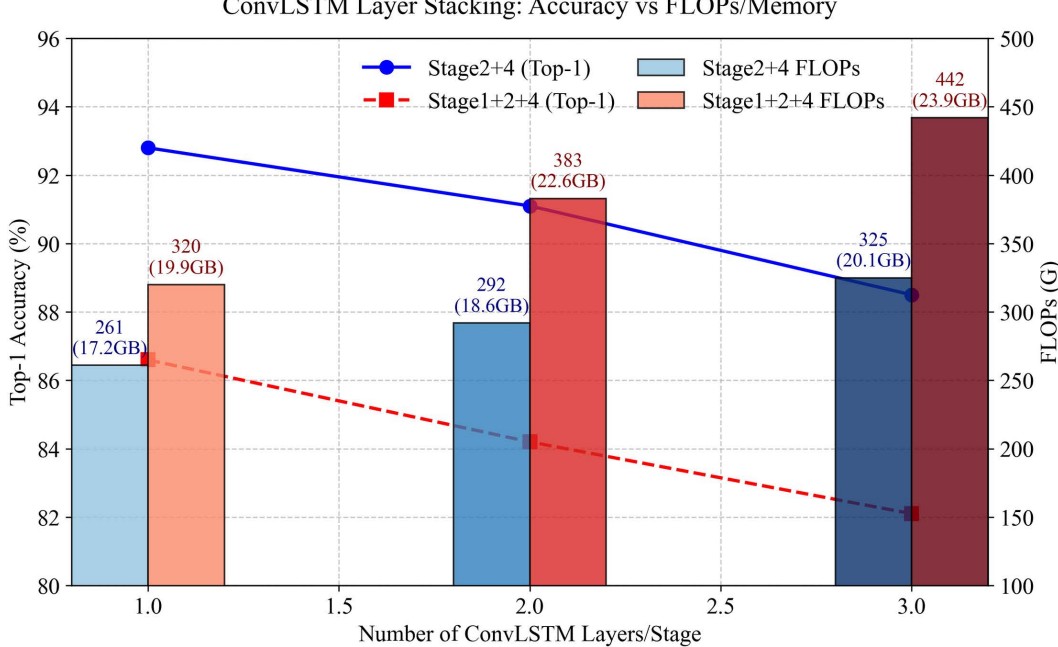

**Fig 7. Accuracy vs FLOPs/Memory Trade-offs in ConvLSTM Layer Stacking.**

**Table 7. An ablation study of whether to use our own pre-trained model on the final experimental results on *UCF-101*.**

| Method | Input | Pretrain | Top-1 (%) | Top-5 (%) | FLOPs (G) |
|---|---|---|---|---|---|
| Video Swin-CLSTM(ours) | RGB | ImageNet-1K | 88.9 | 96.8 | 273G |
| | RGB+Flow | ImageNet-1K | 90.8 | 98.1 | 261G |
| | LessRGB+Flow | ImageNet-1K | 89.8 | 97.7 | 239G |
| Video Swin-CLSTM(ours) | RGB | ImageNet-1K-LSTM | **91.5** | **98.7** | 273G |
| | RGB+Flow | ImageNet-1K-LSTM | **92.8** | **99.4** | 261G |
| | LessRGB+Flow | ImageNet-1K-LSTM | **92.5** | **99.3** | 239G |

accuracy with RGB+optical flow inputs reached 92.8%, compared to 91.5% with RGB inputs alone. This validates the effectiveness of multimodal data fusion.

## Conclusion and future work

This study addresses the challenges of human action recognition in video content analysis by proposing an enhanced Video Swin-CLSTM Transformer model. Through a detailed analysis of existing deep learning methods, we identified performance bottlenecks in recognizing transient actions within complex backgrounds, comprehending long-term semantics, and maintaining computational efficiency. To address these issues, we introduced a fusion strategy that integrates optical flow with spatial information and optimized the model architecture to reduce computational complexity while maintaining high robustness in complex scenarios. Additionally, by embedding the ConvLSTM module into the model, we enhanced its ability to capture long-term semantic information and improved accuracy. Finally, we conducted ablation studies to assess

the impact of various components on model performance. The experimental results reveal that the embedding position, distribution, and stacking layers of the ConvLSTM module, as well as the choice of pre-trained model, significantly influence the final performance. Notably, compared to other Swin Transformer-based methods, our pre-trained model, ImageNet-1K-LSTM, combined with Video Swin-CLSTM, achieved the highest video recognition performance on the UCF-101 dataset.

In future work, we aim to investigate and validate the generalizability of the proposed innovations on more extensive and challenging datasets. Although the proposed model demonstrates robust performance under standard conditions, its effectiveness may diminish under heavy occlusions or extreme lighting due to its reliance on spatially continuous optical flow and fixed-depth temporal modeling. To address these limitations, we plan to investigate the integration of multi-modal inputs (e.g., audio cues for occluded scenes and infrared imaging for low-light conditions) to compensate for compromised visual data. Moreover, we will propose developing adaptive ConvLSTM architectures that dynamically adjust their layer depth based on scene complexity, thereby enabling tailored temporal modeling for a wider range of challenges.

While the ConvLSTM-enhanced model excels at local temporal dynamics, it struggles with long-range semantic integration in multi-modal videos. In the next experiment, we will integrate a learnable external memory module to enhance long-term multi-modal reasoning. This module will maintain dedicated memory banks for various modal data streams within a shared semantic space, enabling cross-modal retrieval through content-based addressing (e.g., using audio queries like "glass breaking" to locate relevant video segments). Adaptive gates will fuse ConvLSTM's local temporal features with global memory states, combining short-term dynamics with persistent semantics. This approach could reduce information loss in extended videos while offering interpretable memory visualization.

In terms of reducing computational overhead, optical flow does lower the cost; however, its approach of pixel-frame-based reduction is inherently limited. Therefore, we plan to incorporate a more efficient tube-based sparse sampling method into our model. By defining tubes of various sizes and depths, the model can learn feature information at different levels in the input video (e.g., fine-grained, short-term actions and coarse-grained, long-term actions), while significantly reducing computational overhead and maintaining a lightweight architecture. These improvements are expected to be implemented and validated in future experiments, ultimately advancing video-based human action recognition and expanding its practical applications across various domains.

## Acknowledgments

The authors thank the organization of Center for Research in Computer Vision (CRCV) for making the datasets available.

## Author contributions

**Conceptualization:** Jun Qin, zheng Ye.

**Data curation:** Shenwei Chen.

**Formal analysis:** Jun Qin.

**Funding acquisition:** Jun Qin, Jing Liu.

**Methodology:** Jun Qin, Shenwei Chen.

**Project administration:** Jing Liu.

**Resources:** Shenwei Chen.

**Software:** Shenwei Chen.

**Supervision:** Jun Qin, zheng Ye.

**Validation:** Shenwei Chen.

**Writing – original draft:** Shenwei Chen, Zhou Liu.

**Writing – review & editing:** Shenwei Chen, zheng Ye.

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
