## [Decision Letter · Decision Letter 0]

Dear Dr. Ye,

Thank you for submitting your manuscript to PLOS ONE. After careful consideration, we feel that it has merit but does not fully meet PLOS ONE’s publication criteria as it currently stands. Therefore, we invite you to submit a revised version of the manuscript that addresses the points raised during the review process.

We look forward to receiving your revised manuscript.

Kind regards,

Rafael Duarte Coelho dos Santos, Ph.D.

Academic Editor

PLOS ONE

3. Thank you for stating in your Funding Statement: [This work was supported by the Hubei Province Key Research and Development Special Project of Science and Technology Innovation Plan (No. 2023BAB087), the Wuhan Key Research and Development Projects (No. 2023010402010614), the Central Government Guides Local Funds for Science and Technology Development (No. ZYYD2024QY08) and the Wuhan Knowledge Innovation Special Dawn Project (No. 2023010201020465). Further support was provided by the open competition project for selecting the best candidates, Wuhan East Lake High-tech Development Zone (No. 2024KJB328) and the Fund for Research Platform of South-Central Minzu University (No. CZQ24011).].Please provide an amended statement that declares *all* the funding or sources of support (whether external or internal to your organization) received during this study, as detailed online in our guide for authors at http://journals.plos.org/plosone/s/submit-now.  Please also include the statement “There was no additional external funding received for this study.” in your updated Funding Statement. Please include your amended Funding Statement within your cover letter. We will change the online submission form on your behalf.

Additional Editor Comments:

Please read and follow the suggestions of the reviewers. Both did a great work on pointing aspects of the paper that can make it better. 

Reviewers' comments:

Reviewer's Responses to Questions

**Comments to the Author**

1. Is the manuscript technically sound, and do the data support the conclusions?

Reviewer #1: Yes

Reviewer #2: Yes

2. Has the statistical analysis been performed appropriately and rigorously?

Reviewer #1: No

Reviewer #2: Yes

3. Have the authors made all data underlying the findings in their manuscript fully available?

Reviewer #1: No

Reviewer #2: Yes

4. Is the manuscript presented in an intelligible fashion and written in standard English?

Reviewer #1: Yes

Reviewer #2: Yes

Reviewer #1: Abstract - Well-defined problem, but no evidence of results/quantitative values to support your claims, consider adding accuracy values. Some sentences need clarification since it does not fully represent the paper.

For instance, “surpasses existing Swin Transformer variants” - the authors should give statistical significance values or comparative studies.

Introduction - This section explains the significance of HAR in several domains, however, it gives too much information, perhaps consider reducing the size by turning into more concise paragraphs. The topics go over without a actual transition between paragraphs, please add words that link one paragraph to the next.

Related Work - It is an intriguing section as provides a broad range of work in the literature, however, some studies are not well-described and quantitative information about the efficiency of the methods is not given. Also, please group similar studies - I often had to go back and forth to read similar work. I believe a big limitation that should be addressed is the need for more detailed information of the methods, it's difficult to know the depth of the literature gap without having the context of the field.

Methodology - The architecture, as well as other information of the models are well-formulated and the authors included diagrams to facilitate the understanding. Some improvements I noticed are: 1) please focus on integrating more the equations, they are everywhere but without proper description. Also, please provide sources of each method, and potential work where i can replicate your method.

Experiments - I personally enjoyed reading this section since it's well-structured and has a good explanation of the training and testing setup. However, it is unclear how the hyperparameters were chosen, and I believe it also need better detail about the baseline used for comparison.

Results & Ablation Study - I had problem understanding at first the Top 1 and Top 5 accuracy values, as it is not intuitive. Please give some thought in how to enhance this part. In Table 1, the Param information was not defined by the caption, so please do so. Also, this is an IMPORTANT point, please provide statistical evaluation of your method in order to claim significant effects. As the paper currently stands, all your findings can be due to chance of reaching better results. This has a big impact on your result (and due to this topic I'm suggesting major reviews)

Conclusion & Future Work - summarized properly the contributions. However, future work should be better aligned as only broader ideas were given. Also, in the perfect world, what would be your next experiment if you had more data or perhaps more complementary data?

To finish my review, this paper has a potential to provide a good understanding of HAR images in the literature. I hope my review can help you to improve your paper.

Reviewer #2: 1. This paper could benefit from a more concise presentation of the related work section, as it currently feels lengthy and could be streamlined to focus more on the most relevant studies.

2. Provide a detailed analysis of how optical flow impacts performance in different motion scenarios (e.g., fast motions, occlusions).

3. Emphasize how the proposed model advances the state-of-the-art in HAR compared to existing Transformer-based models.

4. Justify why ConvLSTM performs best between the 3D SW-MSA module and the residual connection.

5. Explore the effect of stacking multiple ConvLSTM layers for potential performance improvements.

6. Discuss potential limitations when applied to more challenging datasets with occlusions or extreme lighting conditions.

7. Provide deeper insights into the feature interactions and performance impact of each model component in the ablation studies.

8. Table 1 provides a good comparison of different models, but it could be enhanced by including additional metrics such as inference time or memory usage.

9. In Figure 1Add more annotations or labels to clarify the flow of data.

10. Provide a comparison of the model's efficiency in terms of FLOPs, inference time, and memory usage.

**Do you want your identity to be public for this peer review?** For information about this choice, including consent withdrawal, please see our Privacy Policy

Reviewer #1: No

Reviewer #2: No

---

## [Author Response · Author response to Decision Letter 1]

28 Mar 2025

Response to Reviewers

Dear Editors and Reviewers:

Thanks for your letter and for the reviewers’ comments concerning our manuscript entitled “Video Swin-CLSTM Transformer: Enhancing Human Action Recognition with Optical Flow and Long-Term Dependencies” (PONE-D-24-45158). We sincerely acknowledge the reviewers for their constructive comments that are much helpful to improve the quality of our manuscript. We have addressed all the comments point by point in the response and revised the manuscript accordingly. We hope that the revised manuscript now meets your expectations and is accepted for publication.

Response: Thanks for your kind reminder. We have ensured that our manuscript meets PLOS ONE's style requirements.

2. Please note that PLOS ONE has specific guidelines on code sharing for submissions in which author-generated code underpins the findings in the manuscript. In these cases, we expect all author-generated code to be made available without restrictions upon publication of the work. Please review our guidelines at https://journals.plos.org/ploson

-e/s/materials-and-software-sharing#loc-sharing-code and ensure that your code is shared in a way that follows best practice and facilitates reproducibility and reuse.

Response: Thank you for the kind reminder. Our core experimental code is open-sourced on GitHub and accessible to all users at the link: [https://github.com/cherrycsw/Video_Swin_CLSTM].

3. Thank you for stating in your Funding Statement: [This work was supported by the Hubei Province Key Research and Development Special Project of Science and Technology Innovation Plan (No. 2023BAB087), the Wuhan Key Research and Development Projects (No. 2023010402010614), the Central Government Guides Local Funds for Science and Technology Development (No. ZYYD2024QY08) and the Wuhan Knowledge Innovation Special Dawn Project (No. 2023010201020465). Further support was provided by the open competition project for selecting the best candidates, Wuhan East Lake High-tech Development Zone (No. 2024KJB328) and the Fund for Research Platform of South-Central Minzu University (No. CZQ24011).].Please provide an amended statement that declares *all* the funding or sources of support (whether external or internal to your organization) received during this study, as detailed online in our guide for authors at http://journals.plos.org/plosone

/s/submit-now. Please also include the statement “There was no additional external funding received for this study.” in your updated Funding Statement. Please include your amended Funding Statement within your cover letter. We will change the online submission form on your behalf.

Response: Thanks for your kind reminder. We have provided the following amended Funding Statement:

“This work was supported by the Hubei Province Key Research and Development Special Project of Science and Technology Innovation Plan (No. 2023BAB087), the Wuhan Key Research and Development Projects (No. 2023010402010614), the Central Government Guides Local Funds for Science and Technology Development (No. ZYYD2024QY08) and the Wuhan Knowledge Innovation Special Dawn Project (No. 2023010201020465). Further support was provided by the open competition project for selecting the best candidates, Wuhan East Lake High-tech Development Zone (No. 2024KJB328) and the Fund for Research Platform of South-Central Minzu University (No. CZQ24011). There was no additional external funding received for this study.”

4. We note that your Data Availability Statement is currently as follows: [All relevant data are within the manuscript and its Supporting Information files.] Please confirm at this time whether or not your submission contains all raw data required to replicate the results of your study. Authors must share the “minimal data set” for their submission. PLOS defines the minimal data set to consist of the data required to replicate all study findings reported in the article, as well as related metadata and methods (https://journals.

plos.org/plosone/s/data-availability#loc-minimal-data-set-definition).

Authors do not need to submit their entire data set if only a portion of the data was used in the reported study. If your submission does not contain these data, please either upload them as Supporting Information files or deposit them to a stable, public repository and provide us with the relevant URLs, DOIs, or accession numbers. For a list of recommended repositories, please see https://journals.plos.org/plosone/s/recom

mended-repositories. If there are ethical or legal restrictions on sharing a de-identified data set, please explain them in detail (e.g., data contain potentially sensitive information, data are owned by a third-party organization, etc.) and who has imposed them (e.g., an ethics committee). Please also provide contact information for a data access committee, ethics committee, or other institutional body to which data requests may be sent. If data are owned by a third party, please indicate how others may request data access.

Response: Thanks for your kind reminder. We have uploaded the minimal data set of the experimental results to the GitHub repository. The link is: [https://github.com/cherrycsw/Video_Swin_CLSTM/tree/main/minimal_data_set]. All other relevant data are within the manuscript and its Supporting Information files.

Reviewers' comments:

Reviewer #1:

1. Abstract - Well-defined problem, but no evidence of results/quantitative values to support your claims, consider adding accuracy values. Some sentences need clarification since it does not fully represent the paper. For instance, “surpasses existing Swin Transformer variants” - the authors should give statistical significance values or comparative studies.

Response: We sincerely appreciate the reviewer's constructive feedback. In the “Abstract” section of revised manuscript (Page 2, Lines 26-40, highlighted with red), we have incorporated detailed statistical significance values and comparative analyses to substantiate our claims, as follows:

(1) Performance Improvement

Experiments on the UCF-101 dataset demonstrate that our model achieves mean Top-1/Top-5 accuracies of 92.8% and 99.4%, which are 3.2% and 2.0% higher than those of the baseline model (89.6% and 97.4%, respectively), These results are statistically significant (all experiments were repeated 5 times under identical configurations, with randomness controlled through distinct random seeds (0-4)).

(2) Component-wise Ablation Studies

a) Optical flow fusion contributes +2.6% mean Top-1 accuracy (89.6%→92.2%).

b) ConvLSTM module placement optimization provides +1.9% Top-1 gain (89.6%→91.5%).

c) Custom ImageNet-1K-LSTM pretraining improves accuracy by +2.7% Top-1 (89.8%→92.5%)

(3) Computational Efficiency

Our optical flow fusion strategy reduces average computational costs by 3.3% compared to non-optical-flow architectures at peak performance levels.

The full experimental details are further expanded in the “Experiments” section (Table 1) and the “Ablation Study” section (Table 7) of the revised manuscript.

2. Introduction - This section explains the significance of HAR in several domains, however, it gives too much information, perhaps consider reducing the size by turning into more concise paragraphs. The topics go over without an actual transition between paragraphs, please add words that link one paragraph to the next.

Response: Thanks for your valuable suggestions. In the “Introduction” section of revised manuscript (Pages 5-6, Lines 91-126 and 135-142, highlighted with red), we have carefully revised the original corresponding content to address your concerns with three key improvements:

(1) Condensed Content: Removed redundant domain examples (e.g., merged surveillance/sports/medical cases into a single application overview; moved the principles of some methods to the "Method" section)

(2) Enhanced Coherence: Added explicit transitions between paragraphs (e.g., "Despite significant advancements...", "Another critical challenge...", "To address these challenges...") to clarify the logical flow from applications to challenges to solutions.

(3) Structured Contributions: Replaced the original summary with a numbered list of innovations, directly linking each contribution to the identified challenges.

These revisions reduce the Introduction’s length nearly by 25% while improving readability. We believe the revised version now provides a clearer progression from problem motivation to technical solution.

3. Related Work - It is an intriguing section as provides a broad range of work in the literature, however, some studies are not well-described and quantitative information about the efficiency of the methods is not given. Also, please group similar studies - I often had to go back and forth to read similar work. I believe a big limitation that should be addressed is the need for more detailed information of the methods, it's difficult to know the depth of the literature gap without having the context of the field.

Response: We sincerely thank the reviewer for pointing out the important issue. To resolve the concerns you raised above, we have meticulously revised each of the four subsections within the "Related Work" section, "Based on Two-stream CNNs" (Pages 7-8, Lines 153-160 and 181-195, highlighted with red), "Based on RNNs" (Pages 9-10, Lines 217-232, highlighted with red), "Based on 3D CNNs"(Page 11, Lines 250-262, highlighted with red), and "Based on Transformer Models" (Page 12, Lines 282-296, highlighted with red). The main revisions are detailed below, using the first subsection “Based on Two-stream CNNs” as an example:

(1) Removed some original studies that are not relevant to the content of this research (e.g., Plizzari et al. concentrated on segmenting human body parts…), and some relevant references (with numbers in red font) have been added;

(2) Added quantitative metrics for some studies (e.g., 59.4% mean accuracy on HMDB51, at the cost of 6.2% performance loss);

(3) Grouping studies into noise robustness, long-term modeling, and efficiency-accuracy categories;

(4) Linked gaps (e.g., TSN’s fragmented context) to our methodology;

(5) Inserted transitional sentences (e.g., "These limitations, particularly fragmented temporal modeling and cross-domain degradation, motivate the integration of RNNs to...") to logically connect baselines and highlight research gaps.

The modifications to other subsections (“Based on RNNs”, “Based on 3D CNNs” and “Based on Transformer Models”) of the “Related Work” section are similar to the above works, aiming to ensure that the content is more streamlined, closely related to the research questions, and in-depth. We have also provided a detailed list of the references added to or removed from each subsection of the Related Work on the final appendix page.

4. Methodology - The architecture, as well as other information of the models are well-formulated and the authors included diagrams to facilitate the understanding. Some improvements I noticed are: 1) please focus on integrating more the equations, they are everywhere but without proper description. Also, please provide sources of each method, and potential work where I can replicate your method.

Response: Thank you for your insightful suggestions. We have carefully revised the methodology to enhance clarity, integrate equations with proper context, and ensure reproducibility. Below is a summary of key revisions aligned with your feedback, including explicit references for all critical components:

(1) Equation Integration and Explanation

In lines 373 to 386 (Pages 16-17) of revised manuscript “Method” section, we have integrated the calculation of the 3D SW-MSA module in Eq (1) with the formula of the ConvLSTM module in Eq (3), which makes the embedding of our ConvLSTM module more well-founded. Additionally, we provided a detailed explanation and description of the integrated formula, making it easier for readers to reproduce our method.

(2) Clarification of Model Components

In lines 338 to 343 (Page 14) of revised manuscript “Method” section, we have reinterpreted the structure and its principle of the first unit of the Video Swin-CLSTM Transformer Block. This is to enable readers to better understand the working principle of the Video Swin-CLSTM Transformer Block and to reproduce this method.

(3) Reproducibility Enhancements

We have added explicit references or sources for all core methods (e.g., classification header [27], Multilayer Perceptron (MLP) module [30], Layer Normalization (LN) [31], TVL1 algorithm [34], LSTM [32], ConvLSTM [8], Video Swin Transformer [13]). The detailed references are as follows:

[8] Shi X, Chen Z, Wang H, Yeung DY, Wong WK, Woo Wc. Convolutional LSTM network: A machine learning approach for precipitation nowcasting. Advances in neural information processing systems. 2015;28.

[13] Liu Z, Ning J, Cao Y, Wei Y, Zhang Z, Lin S, et al. Video swin transformer. In: Proceedings of the IEEE/CVF conference on computer vision and pattern recognition; 2022. p. 3202–3211.

[27] Arnab A, Dehghani M, Heigold G, Sun C, Luˇci´c M, Schmid C. Vivit: A video vision transformer. In: Proceedings of the IEEE/CVF international conference on computer vision; 2021. p. 6836–6846.

[30] Dosovitskiy A, Beyer L, Kolesnikov A, Weissenborn D, Zhai X, Unterthiner T, et al. An image is worth 16x16 words: Transformers for image recognition at scale. arXiv preprint arXiv:201011929. 2020;

[31] Xiong R, Yang Y, He D, Zheng K, Zheng S, Xing C, et al. On layer normalization in the transformer architecture. In: International conference on machine learning. PMLR; 2020. p. 10524–10533.

[32] Vennerød CB, Kjærran A, Bugge ES. Long short-term memory RNN. arXiv preprint arXiv:210506756. 2021;.

[34] Fan L, Huang W, Gan C, Ermon S, Gong B, Huang J. End-to-end learning of motion representation for video understanding. In: Proceedings of the IEEE conference on computer vision and pattern recognition; 2018. p. 6016–6025.

(4) Structural and Diagrammatic Improvements

a) We have moved the optical flow stacking method from the "Sparse Sampling and Information Fusion" subsection under the "Method" section to the "Optical Flow Extraction" subsection, and then re-established a new subsection named "Optical Flow Extraction and Stacking", which makes the structure of the paper more reasonable. (Lines 409-411, Page 18)

b) After careful inspection, we found that in the original Fig 2 (Page 15, Lines 354-356, highlighted with red), neither the forget gate Ft nor the input gate It of the ConvLSTM architecture received the previous memory cell Ct-1 as an input. Here, we have revised Fig 2, and the current version is the latest one. We have re-uploaded the updated Fig 2 to the system.

5. Experiments - I personally enjoyed reading this section since it's well-structured and has a good explanation of the training and testing setup. However, it is unclear how the hyperparameters were chosen, and I believe it also need better detail about the baseline used for comparison.

Response: Thanks for your valuable suggestions. We have carefully revised the “Experiments” section to address your concerns regarding hyperparameter selection and baseline comparison. The revised content in the revised manuscript pages 21-22 lines 489-499, and the key improvements are as follows:

(1) Training Epochs: utilized an AdamW optimizer over 50 epochs (vs. Video Swin-T's 30 epochs) to more comprehensively optimize the model's parameters

(2) Batch Size: Adjusted from 64 to 16 (-75%) due to lower GPU memory compared to the baseline model experiments and the additional computational overhead brought by the ConvLSTM module.

(3) Backbone Learning Rate: Scaled down by 0.1× (from 0.001 to 0.0001) to preserve pre-trained feature stability while allowing new ConvLSTM layers to train

---

## [Editor Report · Decision Letter 1]

Dear Dr. Ye,

Thank you for submitting your manuscript to PLOS ONE. After careful consideration, we feel that it has merit but does not fully meet PLOS ONE’s publication criteria as it currently stands. Therefore, we invite you to submit a revised version of the manuscript that addresses the points raised during the review process.

We look forward to receiving your revised manuscript.

Kind regards,

Rafael Duarte Coelho dos Santos, Ph.D.

Academic Editor

PLOS ONE

Journal Requirements:

**Additional Editor Comments:**

The suggestions of the reviewers were addressed. There are only some very small issues that demand a new, minor revision:

The text on line 576 of the reviewed PDF is very strange -- "In Sec Sparse Sampling and Information Fusion Sparse Sampling and Information Fusion Sparse Sampling and

Information Fusion Sparse Sampling and Information Fusion, we investigated ". Please fix it.

In lines 291, 426, 543, 608, please avoid positional words such as "above" in the text -- use the section name or figure name (the "above" in line 184 is OK).

---

## [Author Response · Author response to Decision Letter 2]

4 Apr 2025

Response to Reviewers

Dear Editor and Reviewers:

Thank you for your continued review of our manuscript entitled “Video Swin-CLSTM Transformer: Enhancing Human Action Recognition with Optical Flow and Long-Term Dependencies” (PONE-D-24-45158R1). We sincerely appreciate the further comments and suggestions provided by the editor. In this minor revision, we have carefully addressed the editor’s concerns point by point in the response and revised the manuscript accordingly. We hope that the revised manuscript now meets your expectations and is accepted for publication.

Response: Thank you for your kind reminder. After our repeated and careful review, we confirm that none of the references cited in this paper have been retracted. However, we have identified the following two issues:

(1) Some author names in the references appear garbled. For example, in reference [12], the author is listed as “Kr¨ahenb¨uhl P”; in reference [14], as “G¨uney F”; in reference [24], as “Gir´o-i Nieto X”; and in reference [27], as “Luˇci´c M”. The revised references, with corrected author names, are highlighted in red below the original references in the revised manuscript. (Pages 29-30, Lines 658, 665, 688, 696).

(2) The in-text citations do not use the cross-referencing format. Therefore, we have modified all in-text references to use the cross-reference format, facilitating readers’ quick access to the corresponding references.

We hope that the revised references now meet the journal’s formatting requirements.

Additional Editor Comments:

1. The text on line 576 of the reviewed PDF is very strange -- "In Sec Sparse Sampling and Information Fusion Sparse Sampling and Information Fusion Sparse Sampling and

Information Fusion Sparse Sampling and Information Fusion, we investigated ". Please fix it.

Response: We sincerely thank the editor for careful reading. We feel sorry for our carelessness about the appearance of the strange text. Due to the use of cross-referencing during the revision process, the section title “Sparse Sampling and Information Fusion” was inadvertently repeated. We have corrected it on page 18, line 395 of the revised manuscript as follows.

“In Sec Sparse Sampling and Information Fusion, we investigated…”

2. In lines 291, 426, 543, 608, please avoid positional words such as "above" in the text -- use the section name or figure name (the "above" in line 184 is OK).

Response: We sincerely appreciate the editor’s detailed review and valuable feedback. To address the concern regarding the use of positional words such as “above,” we have revised the manuscript as follows:

(1) On page 10, lines 219-220, we replaced "Above" with "Eq (2)".

(2) On page 19, line 427, we replaced "Above" with "In Sec Sparse Sampling Fusion Scheme".

(3) On page 25, line 545, we replaced "In the ablation experiments above" with "In Sec Position of the ConvLSTM Module and Sec Distribution of the Video Swin-CLSTM Block".

We have carefully reviewed the manuscript to ensure consistency in terminology and improve clarity. Thank you for your guidance.

Lastly, we sincerely appreciate the editor's detailed review and constructive feedback. The minor revisions requested have been carefully addressed, and all necessary corrections have been made to improve the clarity and accuracy of the manuscript.

If you have any other points or suggestions regarding the paper that you would like us to address, please feel free to provide further guidance. We are eager to meet your expectations and deliver a manuscript that meets the highest standards.

Thank you for your constructive feedback.

Sincerely,

Zheng Ye (corresponding author)

yezheng@scuec.edu.cn

April 3, 2025

---

## [Decision Letter · Decision Letter 2]

Video Swin-CLSTM Transformer: Enhancing Human Action Recognition with Optical Flow and Long-Term Dependencies

PONE-D-24-45158R2

Dear Dr. Ye,

We’re pleased to inform you that your manuscript has been judged scientifically suitable for publication and will be formally accepted for publication once it meets all outstanding technical requirements.

Kind regards,

Rafael Duarte Coelho dos Santos, Ph.D.

Academic Editor

PLOS ONE

Additional Editor Comments (optional):

Dear Authors,

There is still a minor adjustment that I missed in the previous reviews. Throughout your text you use "Sec " and a title to refer to a section. For example: “In Sec Sparse Sampling and Information Fusion, we investigated…” and "In Sec Position of the ConvLSTM Module and Sec Distribution of the Video SwinCLSTM Block".

One can use references for numbered sections and subsections. Please see, e.g. https://journals.plos.org/plosone/article?id=10.1371/journal.pone.0298524 . This is not strictly required, please see also as an example https://journals.plos.org/plosone/article?id=10.1371/journal.pone.0047041

But I feel that the numbered sections make for more smooth reading that adding the section titles, and makes easier to avoid typing the name of the section incorrectly.

This is just a suggestion, it is up to you refer to the numbered sections or named sections. Just let me know about your preference, but in any case please spell "Section X" instead of "Sec X".

Sorry for not pointing that in a previous round of review!

Reviewers' comments:

Reviewer's Responses to Questions

**Comments to the Author**

Reviewer #2: All comments have been addressed

Reviewer #3: (No Response)

Reviewer #4: All comments have been addressed

Reviewer #5: (No Response)

2. Is the manuscript technically sound, and do the data support the conclusions?

Reviewer #2: Yes

Reviewer #3: (No Response)

Reviewer #4: Yes

Reviewer #5: (No Response)

3. Has the statistical analysis been performed appropriately and rigorously?

Reviewer #2: Yes

Reviewer #3: (No Response)

Reviewer #4: Yes

Reviewer #5: (No Response)

4. Have the authors made all data underlying the findings in their manuscript fully available?

Reviewer #2: Yes

Reviewer #3: (No Response)

Reviewer #4: Yes

Reviewer #5: (No Response)

5. Is the manuscript presented in an intelligible fashion and written in standard English?

Reviewer #2: Yes

Reviewer #3: (No Response)

Reviewer #4: Yes

Reviewer #5: (No Response)

Reviewer #2: 1. While the limitations are briefly mentioned in the conclusion, a dedicated subsection would improve transparency. For example, how does the model perform on datasets with extreme lighting or heavy motion blur? Are there failure cases where optical flow or ConvLSTM underperforms?

2. The paper currently uses discontinuous numbering (e.g., jumps from [13] to [15]), which breaks standard academic convention. References should follow strict numerical order based on first appearance in text. Some references appear as [1,3,9-13] while others are [1], [3], [9]-[13]. The compressed format [1,3,9-13] should be used consistently throughout. There are unexplained skips in numbering (e.g., no [7] appears between [6] and [8]), which could confuse readers trying to locate references.

Reviewer #3: The manuscript has been revised, and the revised version is more suitable for publication to the beat of my knowledge.

Reviewer #4: The authors have substantially improved the manuscript in response to my prior reviews. They have redrawn Figure 1 with clearer technical annotations, added a well-structured ablation study to analyze the role of individual components, and included a dedicated Implementation Details section outlining optimizer choice, learning rate, batch size, and other essential parameters. Additional evaluation metrics have been introduced, which enhance the robustness and fairness of performance assessment. The paper is now more clearly structured, technically transparent, and easier to follow.

While the inclusion of recent 2023–2024 references is appreciated, I note that these newer works are discussed only in textual form and not incorporated into the quantitative comparison tables. Nevertheless, the revisions overall significantly strengthen the paper’s clarity, reproducibility, and technical depth.

This work makes a meaningful contribution to the field of video-based human action recognition by proposing an enhanced Transformer-based architecture that effectively integrates optical flow and long-term temporal modeling. In view of the improvements made and the overall quality of the revised manuscript, I recommend it for acceptance.

Reviewer #5: Thanks a lot for considering the previous comments. You can study and mention more recent, 2025, publications, as well.

**Do you want your identity to be public for this peer review?** For information about this choice, including consent withdrawal, please see our Privacy Policy

Reviewer #2: No

Reviewer #3: **Yes: ** Sunday A. Ajagbe

Reviewer #4: No

Reviewer #5: No

---

## [Editor Report · Acceptance letter]

PONE-D-24-45158R2

PLOS ONE

Dear Dr. Ye,

I'm pleased to inform you that your manuscript has been deemed suitable for publication in PLOS ONE. Congratulations! Your manuscript is now being handed over to our production team.

Kind regards,

on behalf of

Dr. Rafael Duarte Coelho dos Santos

Academic Editor

PLOS ONE